PREPARED FOR SUBMISSION TO JHEP

# Multi-partite entanglement monotones

**Abhijit Gadde**[a], **Shraiyance Jain**[a], **Harshal Kulkarni**[a,b]

[a]*Department of Theoretical Physics*
*Tata Institute for Fundamental Research, Mumbai 400005*
[b]*Indian Institute of Science Education and Research, Kolkata 741246*

*E-mail:* abhijit@theory.tifr.res.in, shraiyance.jain@tifr.res.in,
harshalkulkarni20@gmail.com

ABSTRACT: If we want to transform the quantum of state of a system to another using local processes, what is the probability of success? It turns out that this probability can be bounded by quantifying entanglement within both the states. In this paper, we construct a family of multipartite entanglement measures that are monotonic under local operations and classical communication on average. The measures are constructed out of local unitary invariant polynomials of the state and its conjugate, and hence are easy to compute for pure states. Using these measures we bound the success probability of transforming a given state into another state using local quantum operations and classical communication.

# 1   Introduction and review

Consider a quantum system consisting of multiple quantum systems in distant labs. Let $|\psi\rangle$ be a generic entangled state in this system. Consider the problem of transforming $|\psi\rangle$ to another entangled state $|\phi\rangle$ using quantum operations that are local within each lab, possibly aided by phone calls to other labs. What can be said about the success probability of such a transformation? In particular, can we bound it?

Vidal realized that this question is intimately related to the question of quantifying entanglement [1]. To quantify entanglement, building on the work of [2], he introduced the notion of an *entanglement monotone* $\mu$. We will shortly give a precise definition of the term but essentially it is a function of the state that has following important property:

- It does not increase under local quantum operations and classical communication (`locc`), on average.

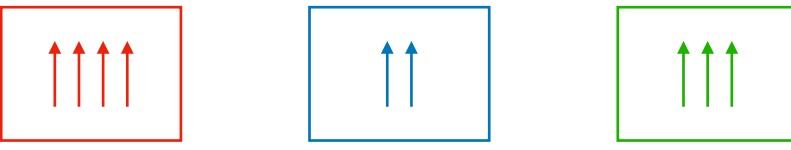

**Figure 1**. Schematic representation of distant labs containing four, two and three qubits respectively.

This property is justified from the point of quantifying entanglement. A general local operation - also known as positive operator valued measurement - can be understood as the effect on the system of an interaction with its local environment. After a general local operation $\Lambda$ on the state $\rho$, we get an ensemble of states with $\rho_i$ appearing with probability $p_i$ (such that $\sum_i p_i = 1$). Each instance of $\rho_i$ corresponds to a particular outcome of the measurement done on the environment.

$$\rho \xrightarrow{\Lambda} \Lambda(\rho) = \{p_i, \rho_i\}, \tag{1.1}$$

$$p_i = \text{Tr} E_i^{(A)} \rho E_i^{(A)\dagger}, \qquad \rho_i = E_i^{(A)} \rho E_i^{(A)\dagger}/p_i. \tag{1.2}$$

Here $E_i^{(A)}$'s are linear transformations on party $A$ which preserve the trace of the density matrix i.e. $E_i^{(A)\dagger} E_i^{(A)} = \mathbb{I}$. They are known as Kraus operators. If one thinks of entanglement as a resource, then it must not increase under local quantum operation. Vidal formalized this property as $\mu_{\text{avg}}(\Lambda(\rho)) \leq \mu(\rho)$ where the average value $\mu_{\text{avg}}$ of the entanglement monotone $\mu$ on the outcome ensemble is defined as the weighted average of entanglement monotone of $\rho_i$ i.e. $\mu_{\text{avg}}(\Lambda(\rho)) = \sum_i p_i \mu(\rho_i)$. It follows that $\mu$ must be invariant under local unitary transformation because such a transformation is an invertible local quantum operation and that $\mu$ is monotonic under local quantum operations.

In addition to performing quantum operations that are local to their respective labs, it is usually assumed that the lab operators can also communicate with each other classically e.g. via a phone call. Classical communication allows one party to communicate the identity of the state of the system in the resulting ensemble to the other parties. However, the parties can choose to dismiss this information. This leads to the treatment of the ensemble $\{p_i, \rho_i\}$ as a mixed state with density matrix $\rho = \sum_i p_i \rho_i$. Vidal thought of this dismissal of information as a type of local operation and because the monotone must not increase under local operations, it must be that $\sum_i p_i \mu(\rho_i) \geq \mu(\sum_i p_i \rho_i)$. In other words, $\mu$ should be convex in $\rho$. To summarize,

---

**Definition 1.1** (Entanglement monotone). *A local unitary invariant function $\mu$ of the density matrix $\rho$ is called an entanglement monotone if the following conditions hold.*

---

*1.* (monotonicity under unilocal operations)

$$\mu_{\mathrm{avg}}(\Lambda(\rho)) \leq \mu(\rho) \qquad \text{where} \qquad \mu_{\mathrm{avg}}(\Lambda(\rho)) \equiv \sum_i p_i \mu(\rho_i), \tag{1.3}$$

$$\rho \xrightarrow{\Lambda} \Lambda(\rho) = \{p_i, \rho_i\}, \qquad p_i = \mathrm{Tr} E_i^{(A)} \rho E_i^{(A)\dagger}, \qquad \rho_i = E_i^{(A)} \rho E_i^{(A)\dagger}/p_i.$$

$E_i^{(A)}$'s are linear transformations on party $A$ which preserve trace i.e. $E_i^{(A)\dagger} E_i^{(A)} = \mathbb{I}$.

2. (convexity) If $\rho = \sum_i p_i \rho_i$ then

$$\mu(\rho) \leq \sum_i p_i \mu(\rho_i). \tag{1.4}$$

3. (normalization) $\mu(\rho) = 0$ if $\rho$ is separable.

Note the distinction between the first two conditions. At first go, the first condition may look like a concavity condition and hence contradicting the second condition of convexity, but that is not so. The first condition is a statement about the value of the monotone after the `locc` operation $\Lambda$ while the second one is simply a convexity condition. In a sense, the first condition is more physical while the second is more mathematical. Also note that the first and the second condition together imply,

$$\mu(\Lambda(\rho)_{\mathrm{mixed\,state}}) \leq \mu(\rho), \qquad \text{where} \quad \Lambda(\rho)_{\mathrm{mixed\,state}} \equiv \sum_i p_i \rho_i \tag{1.5}$$

Here by $\Lambda(\rho)_{\mathrm{mixed\,state}}$ we mean the mixed state corresponding to the ensemble $\{p_i, \rho_i\}$ where $p_i$ and $\rho_i$ are as defined in equation (1.3). This ensures that the monotonicity of $\mu$ holds even under the uncorrelated local operations on multiple parties because they can be thought of as a sequence of unilocal operations (1.3). Let us now comment on the third condition.

**Definition 1.2** (Separable states). *A* `q`*-partite mixed state $\rho$ is called separable if it can be written as*

$$\rho = \sum_i p_i \, \rho_1^{(i)} \otimes \ldots \otimes \rho_{\mathsf{q}}^{(i)}. \tag{1.6}$$

*where $p_i > 0$ and $\rho_i$ are local density matrices.*

Alternatively, separable state is a classical mixture of factorized states. It is only classically correlated and contains no quantum entanglement. An `locc` operation is reversible on the set of separable states. Given that $\mu$ is a monotone, it must be constant on them. Due to the

lack of quantum entanglement, $\mu$ is defined by subtracting this constant so that it vanishes on separable states. Then

$$\mu(\rho) \geq 0 \tag{1.7}$$

for general mixed states $\rho$.

The relevance of entanglement monotones to the problem of bounding the transition probability can be immediately seen thanks to the following theorem.

**Theorem 1.1** (Vidal [1]). *The maximal success probability $p_{\rho \to \rho_*}$ of converting a multi-partite state $\rho$ to $\rho_*$ using* locc *is*

$$p_{\rho \to \rho_*} = \min_{\mu} \frac{\mu(\rho)}{\mu(\rho_*)}. \tag{1.8}$$

*where the minimization is performed over all entanglement monotones $\mu$.*

*Proof.* Let $\rho_*$ be one of the possible outcomes of the positive operator valued measurement on $\rho$. In other words, $\rho_*$ appears as proportional to one of the $E_i \rho E_i^\dagger$, say the one corresponding to $E_*$ and the probability of this outcome is $p_* = \mathrm{Tr}(E_* \rho E_*^\dagger)$. Then, using the properties of entanglement monotones and (1.7),

$$\mu(\rho) \geq p_* \mu(\rho_*) + \sum_{i \in \text{rest}} p_i \mu(\rho_i) \geq p_* \mu(\rho_*). \tag{1.9}$$

This shows that $p_{\rho \to \rho_*} \leq \frac{\mu(\rho)}{\mu(\rho_*)}$ for every $\mu$. As for the equality in equation (1.8), note that $p_{\rho \to \rho_*}$ itself is an entanglement monotone of $\rho$, for a fixed $\rho_*$, and that $p_{\rho_* \to \rho_*} = 1$. $\qquad \square$

Note that if $\mu_i$'s are entanglement monotones then their convex combination $\sum_i p_i \mu_i$ is also an entanglement monotone. In other words, the set of entanglement monotones is convex.

Although, the question of entanglement monotones for bi-partite states has been explored widely in the literature, the same for multi-partite states has received relatively less attention. Some multi-partite entanglement monotones are known in the literature (see [3, 4] for a review) but a general theory is lacking. Our aim in this paper is to remedy this situation and develop a theory of multi-partite monotones of a certain type.

Let us summarize the main results of the paper. We construct an infinite family of multi-partite monotones that are built from local unitary invariant polynomials of the state and its conjugate. These invariant polynomials are characterized by certain type of edge-labeled graphs with q edge-labels where q is the number of parties. In theorem 3.1, we reduce the condition of entanglement monotonicity to a condition on the graph called edge-convexity. We show, in theorem 3.2, that this condition is preserved by the so called cartesian product on graphs. If the number of parties of the graph $\mathcal{Z}_i$ is $\mathsf{q}_i$ then the number of parties of the

cartesian product graph $\mathcal{Z}_1 \square \mathcal{Z}_2$ is $\mathsf{q}_1 + \mathsf{q}_2$. We solve the edge-convexity condition for graphs with $\mathsf{q} = 2$ in lemma 3.6. This result combined with the theorem 3.2 on cartesian product gives us an infinite set of multipartite entanglement monotones as showed in theorem 3.6. Simplest such $\mathsf{q}$-partite monotone is summarized in the corollary 3.1. It is closely related to the so-called $n = 2$ Renyi multi-entropy introduced in [5]. Among the new multi-partite entanglement monotones that we construct in theorem 3.6, the tri-partite one is closely related to the so-called generalized "computable cross-norm". In addition to this, our new results include theorem 2.3 and theorem A.1.

The rest of the paper is organized as follows. In the rest of the section 1, we review the construction of mixed state entanglement as *convex roof extensions* of entanglement monotones that are defined only for pure states. In section 2, we review the intimate relation between pure state entanglement monotones and concave functions of the density matrix obtained by tracing over any one of the parties. We also review the complete set of bi-partite monotones constructed by Vidal and then review the monotones that are known for the multi-partite states. We also prove a structural result about pure state entanglement monotones. Section 3 contains our new results about multi-partite pure state entanglement monotones. We first develop a graph theoretic language to deal with local unitary invariant functions of the state and formulate the condition of concavity as a condition on the associated graph. We solve a necessary condition for concavity (which we call the edge-reflecting condition) completely. We obtain an infinite class of local unitary invariants that are concave and hence are pure state entanglement monotones. Finally, we demonstrate the effectiveness of these new monotones with the help of some examples.

In recent years, ideas related to entanglement have been immensely useful in understanding certain key issues in quantum gravity via the AdS/CFT correspondence. It would be interesting to place our results about $\mathtt{locc}$ monotonicity of multi-partite entanglement in this more physical context. We will not attempt to do so in this paper.

## 1.1 Construction of entanglement monotones

Entanglement monotones are difficult to construct in general. In [1], Vidal gave a strategy to construct monotones for bi-partite systems. This strategy can be generalized to multi-partite systems as well. This strategy is reviewed by M. Horodecki in [6]. There the author offers a viewpoint that is slightly different from Vidal's and is perhaps more direct. The idea is that one starts with a function $\nu(|\psi\rangle)$ that is defined only on *pure states* and then extend it to mixed states in a universal way. Let us first discuss the pure state entanglement monotones. They obey the restriction of the definition 1.1 to pure states i.e. with the density matrices $\rho$ and $\rho_i$ taken to be of rank 1, namely, $|\psi\rangle\langle\psi|$ and $|\psi_i\rangle\langle\psi_i|$ respectively. Written explicitly,

---

**Definition 1.3** (Pure state entanglement monotone (PSEM)). *Let* $|\psi\rangle \rightarrow \Lambda(|\psi\rangle) =$

---

$\{p_i, |\psi_i\rangle\}$. *Then a pure state entanglement $\nu(|\psi\rangle)$ monotone obeys,*

$$\nu(|\psi\rangle) \geq \sum_i p_i \nu(|\psi_i\rangle), \qquad p_i := |E_i^{(A)}|\psi\rangle|^2, \quad \psi_i := E_i^{(A)}|\psi\rangle/\sqrt{p_i}, \tag{1.10}$$

*where $E_i^{(A)}$ are linear operators on any party $A$ that preserve trace.*

They provide a way to bound the transition probabilities from one *pure* state to another. More precisely, the optimal success probability $p_{|\psi\rangle \to |\phi\rangle}$ of converting a multi-partite pure state $|\psi\rangle$ to another multi-partite pure state $|\phi\rangle$ using `locc` is

$$p_{|\psi\rangle \to |\phi\rangle} = \min_\nu \frac{\nu(|\psi\rangle)}{\nu(|\phi\rangle)}. \tag{1.11}$$

where the minimization is performed over the set of `PSEM`s. Note that this set is convex. In fact, it turns out that something stronger is true [4].

**Proposition 1.1** (Szalay[4])**.** *If $G(x_1, \ldots, x_k)$ is a concave and monotonic function of its arguments such that $G(0, \ldots, 0) = 0$, then $G(\nu_1, \ldots, \nu_k)$ is a `PSEM` if all $\nu_i$s are `PSEM`s.*

*Proof.* Let the pure state $|\psi\rangle$ transform into an ensemble of pure states $\Lambda(|\psi\rangle) = \{p_i, |\psi_i\rangle\}$ under the `locc` transformation $\Lambda$.

$$\sum_i p_i G(\nu_1(|\psi_i\rangle), \ldots, \nu_k(|\psi_i\rangle)) \leq G(\sum_i p_i \nu_1(|\psi_i\rangle), \ldots, \sum_i p_i \nu_k(|\psi_i\rangle))$$

$$\leq G(\nu_1(|\psi\rangle), \ldots, \nu_k(|\psi\rangle)). \tag{1.12}$$

In the first line we have used the concavity of $G$ in all of its arguments. In the second line, we have used the monotonicity of $G$, along with the defining `PSEM` property (1.10) of $\nu_i$s. $\square$

**Definition 1.4.** *If a `PSEM` can be written as $G(\nu_1, \ldots, \nu_k)$ where $G$ is a concave and monotonic function of `PSEM`s $\nu_i$ then we call it a composite of $\nu_i$s otherwise we call it a primitive `PSEM`.*

Convex linear combination of $\nu_1, \ldots, \nu_k$ is just one such composite. In particular, this shows that the set of `PSEM`s is convex. Unfortunately, composites can not be used to give better bounds on transition probability. We prove in theorem A.1 that

$$\frac{G(\nu_1(|\psi\rangle), \ldots, \nu_k(|\psi\rangle))}{G(\nu_1(|\phi\rangle), \ldots, \nu_k(|\phi\rangle))} \geq \min\left(\frac{\nu_1(\psi)}{\nu_1(\phi)}, \ldots, \frac{\nu_k(\psi)}{\nu_k(\phi)}, 1\right). \tag{1.13}$$

This makes it clear that we do not need to explore the bounds on transition probabilities obtained from composites of PSEMs constructed in section 3.

This discussion also motivates the definition of a complete set.

> **Definition 1.5.** *A set $\mathcal{K}$ of PSEMs is called complete if*
>
> $$p_{|\psi\rangle \to |\phi\rangle} = \min_{\mu \in \mathcal{K}} \frac{\nu(|\psi\rangle)}{\nu(|\phi\rangle)} \qquad \forall \, |\psi\rangle, |\phi\rangle. \tag{1.14}$$
>
> *A set that is a subset of all complete sets is called the minimal complete set.*

It is clear that all the primitive PSEMs form a complete set and that the PSEMs in a minimal complete set must be primitive. For bi-partite pure states, Vidal computed the optimal probability by constructing a *minimal complete* set of PSEMs. We will review this in section 2.1. Characterizing the minimal set for multi-partite pure states is a difficult exercise, one which we will not undertake in this paper.

## 1.2 Pure states to mixed states

In this subsection we discuss the extension of pure state entanglement monotones to mixed states. It is not essential to the main point of the paper and is added here for completeness. The reader can skip this subsection in the first read.

A pure state entanglement monotone is extended to mixed states using the so called *convex roof extension* [7]. We first write the given mixed state as an ensemble of pure states $\rho = \sum_i p_i |\psi_i\rangle\langle\psi_i|$. There are multiple pure-state ensembles that realize $\rho$. Let us denote their set as $\mathcal{E}_\rho$. The convex roof function $\mathtt{CR}_\nu(\rho)$ on mixed states is defined as,

> **Definition 1.6** (Convex roof).
>
> $$\mathtt{CR}_\nu(\rho) := \min_{\mathcal{E}_\rho} \sum_i p_i \, \nu(|\psi_i\rangle), \qquad \rho = \sum_i p_i |\psi_i\rangle\langle\psi_i|. \tag{1.15}$$

For future use, let us denote the pure state ensemble realizing the minimum as $\mathtt{e}_\nu(\rho)$. The convex roof extension of the entanglement entropy was one of the first mixed state entanglement measures that was constructed and studied [8]. This measure was termed the entanglement of formation.

> **Lemma 1.1** (Horodecki [6]). $\mathtt{CR}_\nu(\rho)$ *is a mixed state entanglement monotone.*

*Proof.* First note that by virtue of being a convex roof $\mathtt{CR}_\nu(\rho)$ is convex. This is because the union of sets $\mathcal{E}_{\rho_1}$ and $\mathcal{E}_{\rho_2}$ over which we do minimization to compute $p\mathtt{CR}_\nu(\rho_1)+(1-p)\mathtt{CR}_\nu(\rho_2)$ is a subset of $\mathcal{E}_{p\rho_1+(1-p)\rho_2}$.

Now we will show that $\mathtt{CR}_\nu(\rho)$ obeys the equation (1.3). Let $\rho$ be an ensemble of pure states as $\rho = \sum_i p_i |\psi_i\rangle\langle\psi_i|$. Its transformation with respect to the set $E_k$ produces

$$\Lambda(\rho) = \sum_k q_k \sigma_k, \qquad q_k := \mathrm{Tr}(E_k \rho E_k^\dagger), \qquad \sigma_k := E_k \rho E_k^\dagger / q_k. \tag{1.16}$$

Using the expression for $\rho$ in terms of pure states,

$$\sigma_k = \frac{1}{q_k} \sum_i p_i q_k^i |\psi_i^k\rangle\langle\psi_i^k|, \qquad q_k^i := \mathrm{Tr}(E_k |\psi_i\rangle\langle\psi_i| E_k^\dagger), \qquad |\psi_i^k\rangle := E_k |\psi_i\rangle / \sqrt{q_i^k}. \tag{1.17}$$

We want to show that $\mathtt{CR}_\nu(\rho) \geq \sum_k q_k \mathtt{CR}_\nu(\sigma_k)$. As $\sigma_k$ is given as the convex combination as in equation (1.17), using convexity of $\mathtt{CR}_\nu$,

$$\sum_k q_k \mathtt{CR}_\nu(\sigma_k) \leq \sum_k \sum_i p_i q_k^i \nu(|\psi_i^k\rangle) \leq \sum_i p_i \nu(|\psi_i\rangle). \tag{1.18}$$

In the last inequality, we have used the pure state monotonicity (1.10). As this equation is true for any pure state ensemble realizing $\rho$, it is also true for the optimal pure state ensemble $\mathtt{e}_\nu(\rho)$. Taking the optimal ensemble, the right-hand side of the inequality (1.18) is precisely $\mathtt{CR}_\nu(\rho)$. $\qquad\square$

In general, computing the convex roof is a difficult task. See [9–11] for various methods. In the rest of the paper, we will not discuss the extension of $\nu$ to mixed states via convex roof construction but rather restrict ourselves to asking the question about maximal success probability $p_{|\psi\rangle\to|\phi\rangle}$ of converting a pure state $|\psi\rangle$ to another pure state $|\phi\rangle$. It is given by

$$p_{|\psi\rangle\to|\phi\rangle} = \min_\nu \frac{\nu(|\psi\rangle)}{\nu(|\phi\rangle)}. \tag{1.19}$$

Here the minimization is performed over all the PSEMs $\nu$. As argued earlier, it suffices to minimize only over primitive PSEMs.

Before we make the PSEMs the primary focus of the paper, let us comment that there are mixed state entanglement monotones that are not defined using convex roofs. They include an appropriately defined distance of $\rho$ from the set of separable states e.g. [2, 12]. Just like convex roofs, these measures are also difficult to compute. Finally, in the bi-partite case, there is a monotone that is efficiently computable - even for mixed states - known as logarithmic negativity [13].

## 2   Pure state entanglement monotones

Vidal gave a simple and elegant way of constructing PSEMs. Consider a local unitary invariant function of q-partite pure state $|\psi\rangle$. Because of local unitary invariance, it can be equivalently

thought of as a function of $\rho_{\bar{A}}$ which is the $(\mathsf{q}-1)$-partite density matrix obtained from $|\psi\rangle$ by tracing out party $A$, for any party $A$. By abuse of notation, we will denote such a function either as $f(|\psi\rangle)$ or as $f(\rho_{\bar{A}})$, according to convenience.

**Theorem 2.1** (Vidal [1]). *A local unitary invariant function $f(|\psi\rangle)$ that is concave in the partial trace $\rho_{\bar{A}} := \mathrm{Tr}_A|\psi\rangle\langle\psi|$ for all parties $A$ is a* PSEM.

Before we present the proof of this theorem, let us understand how to interpret the function $f(|\psi\rangle)$ of the state as a function of the partially traced density matrix $\rho_{\bar{A}}$. With slight abuse of notation, we will denote this function also as $f(\rho_{\bar{A}})$. Given a $\rho_{\bar{A}}$ on $\mathsf{q}-1$ parties, we purify it to a pure state $|\tilde{\psi}\rangle$ on $\mathsf{q}$ parties by essentially reintroducing the traced out party $A$. Let us denote the new party as $\tilde{A}$. The dimension of $H_{\tilde{A}}$ needs to be at least as large as the rank of $\rho_{\bar{A}}$ but it can be larger. It is well-known that all such purifications are related to each other by a local unitary transformation on $\tilde{A}$. Now we define $f(\rho_{\bar{A}}) = f(|\tilde{\psi}\rangle)$ and because $f(|\psi\rangle)$ is a local unitary invariant function, $f(\rho_{\bar{A}})$ doesn't depend on the purification and hence is uniquely defined.

*Proof.* Let the state $\rho = |\psi\rangle\langle\psi|$ be converted to

$$\Lambda(\rho) = \sum_i p_i |\psi_i\rangle\langle\psi_i|, \qquad p_i := |E_i|\psi\rangle|^2, \quad \psi_i := E_i|\psi\rangle/\sqrt{p_i} \tag{2.1}$$

after a local operation by party $A$. We need to show that

$$f(|\psi\rangle) \geq \sum_i p_i f(|\psi_i\rangle). \tag{2.2}$$

The operation (2.1) keeps $\rho_{\bar{A}}$ invariant i.e. $\Lambda(\rho)_A = \rho_{\bar{A}}$ because of the trace preserving property $\sum_i E_i^{(A)\dagger} E_i^{(A)} = \mathbb{I}$. So we have

$$\rho_{\bar{A}} = \sum_i p_i \rho_{i,\bar{A}}, \qquad \rho_{i,\bar{A}} := \mathrm{Tr}_A|\psi_i\rangle\langle\psi_i|. \tag{2.3}$$

The concavity of $f$ in $\rho_{\bar{A}}$ implies, $f(\rho_{\bar{A}}) \geq \sum_i p_i f(\rho_{i,\bar{A}})$. When expressed in terms of purifications $|\psi\rangle$ and $|\psi_i\rangle$'s, this equation is exactly the one that we want to prove. Note that, for this argument to go through, it is necessary that the density matrices $\rho_{\bar{A}}$ and $\rho_{i,\bar{A}}$ are obtained by tracing out precisely one party and not more. As the concavity holds for partial trace $\rho_{\bar{A}}$ for any party $A$, the monotonicity also holds for local operations by any party. $\square$

The main point of the paper is to construct PSEMs for multi-partite states using the theorem 2.1.

## 2.1 For bi-partite states

Because the local unitary invariant data in the partial trace $\rho_{\bar{A}}$ of a bi-partite state are its eigenvalues $\lambda_i$ and because $\lambda_i$ are homogeneous in $\rho_{\bar{A}}$, the PSEMs are concave functions of $\lambda_i$, thanks to theorem 2.1. Also note that the eigenvalues $\lambda_i$ are symmetric with respect to both parties as required by the theorem 2.1. As it turns out, one does not need to minimize over all such functions in (1.19) to get the optimal success probability. In [14], Vidal identified a complete set of PSEMs that is finite. Arranging the nonzero eigenvalues of $\rho_{\bar{A}}$ in descending order, $\lambda_1 \geq \lambda_2 \geq \ldots$, define

$$\tilde{\nu}_k^V = 1 - \sum_{i=1}^{k} \lambda_i. \tag{2.4}$$

As shown in theorem 2.2, the functions $\tilde{\nu}_k^V$ are PSEMs. Moreover, the set $\mathcal{K} = \{\tilde{\nu}_k^V : k = 1, \ldots, d\}$, where $d$ is the number of nonzero eigenvalues of $\rho_{\bar{A}}$, is complete. Vidal proved the completeness of $\mathcal{K}$ by giving an explicit conversion protocol $|\psi\rangle \to |\phi\rangle$ that realizes the success probability obtained by minimizing over $\mathcal{K}$.

**Theorem 2.2** (Vidal [14])**.** $\tilde{\nu}_k^V$ *are* PSEM*s.*

*Proof.* We will show that $\tilde{\nu}_k^V$ are concave function of the density matrix $\rho_{\bar{A}}$. This, in conjunction with theorem 2.1 proves that $\tilde{\nu}_k^V$ are PSEMs. It is convenient to use the following characterization of $\sum_{i=1}^{k} \lambda_i$.

$$\sum_{i=1}^{k} \lambda_i = \max_{P_k} \text{Tr}(P_k \, \rho_{\bar{A}}), \tag{2.5}$$

where $P_k$ are rank $k$ projectors and maximization is performed over all such projectors. This fact is known as Ky Fan's maximum principle. Clearly,

$$\max_{P_k} \text{Tr} P_k \, (p\rho_{1,\bar{A}} + (1-p)\rho_{2,\bar{A}}) \leq p \max_{P_k} \text{Tr}(P_k \, \rho_{1,\bar{A}}) + (1-p) \max_{P_k} \text{Tr}(P_k \, \rho_{2,\bar{A}}). \tag{2.6}$$

This is because, the right-hand side allows for two different choices of projectors to maximize the two terms but on the left-hand side, the same projector has to be used for both terms. This shows that $\sum_{i=1}^{k} \lambda_i$ is a convex function of $\rho$ and hence $\tilde{\nu}_k^V$ is a concave function of $\rho$. The constant 1 is chosen so that it vanishes on the factorized state i.e. for rank 1 density matrix state with $\lambda_1 = 1$ and the rest of the $\lambda$'s zero. $\square$

Note that when

$$\min_{k} \frac{\tilde{\nu}_k^V(|\psi\rangle)}{\tilde{\nu}_k^V(|\phi\rangle)} \geq 1, \tag{2.7}$$

Vidal's theorem states that it is possible to convert $|\psi\rangle$ to $|\phi\rangle$ with unit probability. This agrees beautifully with Nielsen's result [15] that a complete conversion of $|\psi\rangle$ to $|\phi\rangle$ is possible using locc if and only if eigenvalues of $\rho_{\bar{A}}(|\psi\rangle)$ majorize eigenvalues of $\rho_{\bar{A}}(|\phi\rangle)$. The condition (2.7) is simply a rewriting of Nielsen's majorization condition.

It is useful to write an expression for $\tilde{\nu}_k^V$ that is manifestly symmetric in both parties. For this, we note that

$$\max_{P_{k_1,k_2}} |P_{k_1,k_2}|\psi\rangle|^2 = \sum_{i=1}^{\min(k_1,k_2)} \lambda_i. \tag{2.8}$$

Here $P_{k_1,k_2} := P_{k_1} \otimes P_{k_2}$ where $P_{k_1}$ and $P_{k_2}$ are the rank $k_1$ and rank $k_2$ projectors on both the parties respectively. So we define another form of the same monotones,

$$\nu_{k_1,k_2}^V := 1 - \max_{P_{k_1,k_2}} |P_{k_1,k_2}|\psi\rangle|^2 = \tilde{\nu}_{\min(k_1,k_2)}^V. \tag{2.9}$$

Here we have introduced redundancy by writing the monotone as a function of two integers $k_1$ and $k_2$ although it is only a function of $\min(k_1,k_2)$.

## 2.2 For multi-partite states

Before we start constructing PSEMs for multi-partite states, let us look at the problem of conversion $|\psi\rangle \to |\phi\rangle$. In general, this conversion is possible only if $|\psi\rangle\langle\psi|$ is related to $|\phi\rangle\langle\phi|$ by a stochastic version of local operation and classical communication (slocc) [16]. Mathematically an slocc is expressed as,

$$|\psi\rangle\langle\psi| \to |\phi\rangle\langle\phi| \equiv M_1 \otimes \ldots \otimes M_{\mathsf{q}} |\psi\rangle\langle\psi| M_1^\dagger \otimes \ldots \otimes M_{\mathsf{q}}^\dagger|_{\text{normalized}}, \qquad M_A^\dagger M_A \preceq \mathbb{I}. \tag{2.10}$$

The notation $A \preceq \mathbb{I}$ implies that the matrix $\mathbb{I} - A$ is positive definite. Indeed this has to be the case if $|\phi\rangle$ were to appear as one of the possible mixed states under local operations. The reason for the condition $M_A^\dagger M_A \preceq \mathbb{I}$ is that $M_A$ belongs to a set of Kraus operators, in this case a pair, the other being $\tilde{M}_A \equiv \sqrt{\mathbb{I} - M_A^\dagger M_A}$. In the bi-partite case, any two density matrices (of equal rank) are related to each other in this way but that is not always the case for multi-partite density matrices or even for multi-partite pure states. If they are not in the same slocc orbit then the success probability of converting either $|\psi\rangle \to |\phi\rangle$ or $|\phi\rangle \to |\psi\rangle$ is zero. Hence the problem of bounding conversion success only makes sense if $|\psi\rangle$ and $|\phi\rangle$ are in the same slocc orbit i.e. if $|\phi\rangle = M_1 \otimes \ldots \otimes M_{\mathsf{q}} |\psi\rangle$. In the rest of the paper, we will assume that that is the case. If two states belong to the same slocc orbit can be checked by computing quantities that are invariant under local special linear transformations. The are constructed by contracting the indices of $|\psi\rangle$ using the invariant $\epsilon$ tensor for each party.

This discussion also suggests a particular protocol of transformation $|\psi \to |\phi\rangle$ thereby giving a lower bound on the transformation probability. The protocol consists of the action of a pair of Kraus operators $M_A, \tilde{M}_A$ each party $A$. In this case, $|\phi\rangle\langle\phi|$ definitely appears as

one of the $2^q$ terms in the ensemble (it may also appear elsewhere in the sum as well). Its coefficient is

$$p_* = \text{Tr} M_1 \otimes \ldots \otimes M_q |\psi\rangle\langle\psi| M_1^\dagger \otimes \ldots \otimes M_q^\dagger. \tag{2.11}$$

This gives us a lower bound on $p_{|\psi\rangle \to |\phi\rangle}$. In order to get the best lower bound for this protocol, we take all the $M_A$'s to be normalized such that their maximum singular value is 1. This allows the inequality $M_A^\dagger M_A \preceq \mathbb{I}$ to be satisfied while giving the best lower bound for this protocol. Of course, there might exist other protocols which allow for a more efficient conversion of $|\psi\rangle$ to $|\phi\rangle$. Below we give some examples of PSEMs for multi-partite states.

## PSEM of [17]

In [17], the authors showed that any linearly homogeneous positive function of $\rho = |\psi\rangle\langle\psi|$ that is invariant under local special linear transformations i.e. under transformation

$$|\psi\rangle\langle\psi| \to |\phi\rangle\langle\phi| \equiv M_1 \otimes \ldots \otimes M_q |\psi\rangle\langle\psi| M_1^\dagger \otimes \ldots \otimes M_q^\dagger, \qquad \text{Det } M_A = 1. \tag{2.12}$$

is a PSEM. Because of this invariance, for a given slocc orbit the expression for the monotone simplifies to

$$\nu_{\text{Det}}(|\psi\rangle) = \kappa \langle\psi|\psi\rangle^{-1} \tag{2.13}$$

where $|\psi\rangle$ is an un-normalized state obtained from some reference state $|\psi_0\rangle$ in the same slocc orbit by determinant 1 operations and $\kappa$ is a constant that only depends on the orbit.

## PSEM of [18]

In [18] the authors defined the Schmidt measure (SM) of a multi-partite state,

**Definition 2.1** (Schmidt measure). *SM is the $\log_2$ of the minimum value of $k$ such that*

$$|\psi\rangle = \sum_{i=1}^{k} c_i |\psi\rangle_1^{(i)} \otimes \ldots \otimes |\psi\rangle_q^{(i)}. \tag{2.14}$$

They showed that SM is a PSEM.

## PSEM of [19]

A natural generalization of Vidal's monotones for multi-partite case was constructed in [19] using the Ky Fan's maximum formulation as presented in equation (2.9). Consider

$$\nu_{k_1,\ldots,k_q}^{BL} := 1 - \max_{P_{k_1,\ldots,k_q}} |P_{k_1,\ldots,k_q}|\psi\rangle|^2, \qquad P_{k_1,\ldots,k_q} = P_{k_1} \otimes \ldots \otimes P_{k_q}. \tag{2.15}$$

It was shown in [19] that $\nu_{k_1,\ldots,k_q}^{BL}$ are multi-partite PSEMs. We will present a different proof of the same as a special case of a general construction of PSEMs.

## 2.3 A general construction

In this subsection, we will construct a new PSEM from a given PSEM that is not a composite. We will use Vidal's theorem 2.1 to do so. Our starting point for this construction is a linearly homogenous convex function of the partial trace $\rho_{\bar{A}}$. In particular, it is defined for un-normalized states.

**Theorem 2.3.** *Let $f(\rho_{\bar{A}})$ be a linearly homogeneous convex function of $\rho_{\bar{A}}$ that is invariant under local unitary transformations. Then the function*

$$\tilde{f}_{k_1,\ldots k_q}(|\psi\rangle) \equiv \max_{P_{k_1,\ldots,k_q}} f(P_{k_1,\ldots,k_a}|\psi\rangle) \tag{2.16}$$

*also is a linearly homogeneous convex function of $\rho_{\bar{A}}$ that is invariant under local unitary transformations.*

*Proof.* Recall that $f(\rho_{\bar{A}})$ can also be written as a function on the pure states $f(|\psi\rangle)$ where $|\psi\rangle$ is a purification of $\rho_{\bar{A}}$. Convexity of $f(\rho_{\bar{A}})$ means that it obeys

$$f(p\rho_{\bar{A}}^{(1)} + (1-p)\rho_{\bar{A}}^{(2)}) \leq pf(\rho_{\bar{A}}^{(1)}) + (1-p)f(\rho_{\bar{A}}^{(2)}), \tag{2.17}$$

where $\rho_{\bar{A}}^{(i)}$ are normalized density matrix. Thanks to linear homogeneity of $f(\rho_{\bar{A}})$, the inequality continues to hold even if we take $\rho_{\bar{A}}^{(i)}$'s to be un-normalized. Lets prove it. Taking $\rho_{\bar{A}}^{(i)}$ to be un-normalized, define its normalized version as $\rho_{\bar{A}}^{(i),N} \equiv \rho_{\bar{A}}^{(i)}/n_i$ where $n_i = \mathrm{Tr}\rho_{\bar{A}}^{(i)}$.

$$\begin{aligned} f(p\rho_{\bar{A}}^{(1)} + (1-p)\rho_{\bar{A}}^{(2)}) &= f(n(p\frac{n_1}{n}\rho_{\bar{A}}^{(1),N} + (1-p)\frac{n_2}{n}\rho_{\bar{A}}^{(2),N})) \\ &= nf(p\frac{n_1}{n}\rho_{\bar{A}}^{(1),N} + (1-p)\frac{n_2}{n}\rho_{\bar{A}}^{(2),N}) \end{aligned} \tag{2.18}$$

Here $n$ is $\mathrm{Tr}(p\rho_{\bar{A}}^{(1)} + (1-p)\rho_{\bar{A}}^{(2)})$. In the second line we have used linearity of $f$. Now we have $f$ evaluated on the normalized density matrix that is expressed as convex linear combination of two normalized density matrices.

$$\begin{aligned} f(p\rho_{\bar{A}}^{(1)} + (1-p)\rho_{\bar{A}}^{(2)}) &\leq n\left(p\frac{n_1}{n}f(\rho_{\bar{A}}^{(1),N}) + (1-p)\frac{n_2}{n}f(\rho_{\bar{A}}^{(2),N})\right) \\ &= pf(\rho_{\bar{A}}^{(1)}) + (1-p)f(\rho_{\bar{A}}^{(2)}). \end{aligned} \tag{2.19}$$

In the first line we used the convexity of $f$ for normalized density matrices and in the second line we again used its linearity.

Now we move on to show

$$\tilde{f}_{k_1,\ldots,k_q}(p\rho_{\bar{A}}^{(1)} + (1-p)\rho_{\bar{A}}^{(2)}) \leq p\tilde{f}_{k_1,\ldots,k_q}(\rho_{\bar{A}}^{(1)}) + (1-p)\tilde{f}_{k_1,\ldots,k_q}(\rho_{\bar{A}}^{(2)}), \tag{2.20}$$

where $\rho_{\bar{A}}^{(i)}$ are normalized density matrices. Let $|\psi_A^{(i)}\rangle$ be their purifications. Recall,

$$\tilde{f}_{k_1,\ldots,k_\mathsf{q}}(p\rho_{\bar{A}}^{(1)} + (1-p)\rho_{\bar{A}}^{(2)}) \equiv \max_{P_{k_1},\ldots,P_{k_\mathsf{q}}} f(P_{k_1,\ldots,P_{k_\mathsf{q}}}(\sqrt{p}|\psi^{(1)}\rangle + \sqrt{1-p}|\psi^{(2)}\rangle)). \qquad (2.21)$$

Let $P_{k_1,\ldots,k_\mathsf{q}}^*$ be the projector that maximizes the right hand side. Let $P_{k_1,\ldots,k_\mathsf{q}}^*|\psi^{(i)}\rangle = |\tilde{\psi}^{(i)}\rangle$ be two un-normalized states. Then

$$\tilde{f}_{k_1,\ldots,k_\mathsf{q}}(p\rho_{\bar{A}}^{(1)} + (1-p)\rho_{\bar{A}}^{(2)}) = f(\sqrt{p}|\tilde{\psi}^{(1)}\rangle + \sqrt{1-p}|\tilde{\psi}^{(2)}\rangle). \qquad (2.22)$$

Using the convexity of $f$ for un-normalized states and linearity,

$$\tilde{f}_{k_1,\ldots,k_\mathsf{q}}(p\rho_{\bar{A}}^{(1)} + (1-p)\rho_{\bar{A}}^{(2)}) \leq pf(|\tilde{\psi}^{(1)}\rangle) + (1-p)f(|\tilde{\psi}^{(2)}\rangle). \qquad (2.23)$$

Clearly, $f(|\tilde{\psi}^{(i)}\rangle) \leq \max_{P_{k_1},\ldots,k_\mathsf{q}} f(P_{k_1,\ldots,k_\mathsf{q}}|\psi^{(i)}\rangle)$. This shows that the right hand side of the above inequality is $\leq p\tilde{f}(|\psi^{(1)}\rangle) + (1-p)\tilde{f}(|\psi^{(2)}\rangle)$. $\qquad\qquad\square$

If we take $f(\rho_{\bar{A}}) = \text{Tr}\rho_{\bar{A}}$. Then the above proof shows that the measure (2.15) introduced in [19] is indeed a PSEM.

One thing that is common in almost all the multi-partite PSEMs is that they are all difficult to compute. Given their operational significance, one would like to construct PSEMs that are easy to compute. Also, let us also stress that unlike the bi-partite case, the set of multi-partite PSEMs is very difficult to characterize in general. The question of identifying a minimal complete set of PSEMs or even whether such a non-trivial minimal complete set exists, we believe, is extremely difficult to answer. Instead, in the rest of the paper, we will construct new families of such quantities. Interestingly, they will be constructed from convex functions of $\rho_{\bar{A}}$ that are linearly homogenous, hence they are amenable to the generalization discussed in above theorem 2.3.

For completeness let us comment that, unlike bi-partite case, converting a multi-partite state $|\psi\rangle$ to state $|\phi\rangle$ with unit probability using locc is almost impossible. When it is possible, a criterion involving the so-called "associate density matrices" was obtained in [20]. We will not discuss those results here.

## 3 PSEMs from polynomials in state

In this section, we will construct a set of PSEMs for multi-partite states from local unitary invariant polynomials of the wave-function and its complex conjugate. In particular, we will construct the local unitary invariants that are convex in partially traced density matrix $\rho_{\bar{A}}$ for each party $A$. They will lead us to PSEMs thanks to theorem 2.1.

It is useful to introduce a graphical notation to describe such invariants. After picking a basis $|i_A\rangle$, $A = 1,\ldots,d_A$ for Hilbert space $H_A$, the state in the tensor product $\bigotimes_A H_A$ is written as

$$|\psi\rangle = \sum \psi_{i_1,\ldots,i_\mathsf{q}} |i_1\rangle \otimes \ldots \otimes |i_\mathsf{q}\rangle. \qquad (3.1)$$

The components $\psi_{i_1,\ldots,i_q}$ is the wavefunction of the state $|\psi\rangle$ in the chosen basis. Invariants of local unitary transformations are constructed by taking, say $r$ copies of $\psi$ and $r$ copies of $\bar{\psi}$ and contracting fundamental indices of $\psi$'s with anti-fundamental indices of $\bar{\psi}$'s. It is convenient to use a graphical notation to describe such invariants. Let us denote a state $\psi_{i_1,\ldots,i_q}$ (its complex conjugate $\bar{\psi}_{i_1,\ldots,i_q}$) as a white (black) q-valent vertex. Each edge had a label of one of the q-parties. The edge corresponding to party $A$ is called an $A$-edge and so on. This notation is illustrated in figure 2. In the figures, we denote the edge label using a color that is not black or white.

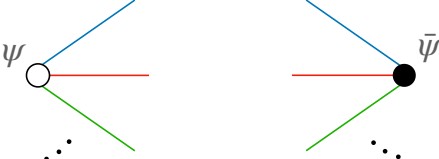

**Figure 2**. White (black) vertex denoting $\psi$ ($\bar{\psi}$). The parties are labeled by colored edges.

Using this graphical notation, it is straightforward to describe local unitary invariants. Whenever an index $i_A$ of a pair of $\psi$ and $\bar{\psi}$ is contracted, we connect the two corresponding vertex with $A$-edge and so on. If all the edges are contracted, the graph represents a local unitary invariant and if some edges are left unconnected then the open graph represents a tensor that transforms appropriately under local unitary transformations. We call the graph of $\psi$, $\bar{\psi}$ vertices representing a local unitary invariant a $\psi$-*graph*. Figure 3 shows an example of a $\psi$-graph. Graph theoretically, it is a q-regular, q-edge-colorable, bi-partite graph for

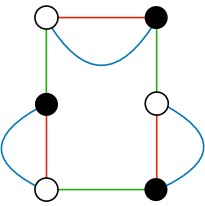

**Figure 3**. Example of a $\psi$-graph constructed from three copies of $\psi$ and $\bar{\psi}$ each by connecting edges of identical colors.

some q. We use the same calligraphic letter, such as $\mathcal{Z}$, to denote the $\psi$-graph as well as the corresponding local unitary invariant function of $\psi$.

**Proposition 3.1.** *If a $\psi$-graph $\mathcal{Z}$ is isomorphic to itself after vertex parity flip then $\mathcal{Z}$ is real.*

This is because vertex parity flip swaps $\psi$ with $\bar{\psi}$ and hence corresponds to complex conjugation. It is also possible to characterize positivity of $\mathcal{Z}$ graph-theoretically. Let us first recall that a graph cut of a connected graph is a subset of edges after removing which, the graph becomes disconnected. From now on, without loss of generality, we will assume $\mathcal{Z}$ is connected.

---

**Definition 3.1.** *A graph cut of a $\psi$-graph is called reflecting if it cuts the graph into two graphs that are isomorphic as edge-labeled graphs with isomorphism flipping vertex parity and if the cut edges connect only those vertices that are images of each other under the said isomorphism.*

---

**Proposition 3.2.** *If a $\psi$-graph $\mathcal{Z}$ admits a reflecting cut then $\mathcal{Z}$ is positive.*

---

*Proof.* Consider the two graphs obtained after the reflecting cut. Restoring the cut edges on each of them separately but not joining them gives us a pair of open graphs that represents tensors $|\mathcal{T}\rangle$ and $|\bar{\mathcal{T}}\rangle$ that are complex conjugates of each other. The original $\psi$-graph $\mathcal{Z}$ is obtained by connecting these indices appropriately. This gives the presentation of $\mathcal{Z}$ as the squared norm of $|\mathcal{T}\rangle$. Hence $\mathcal{Z}$ is positive. $\qquad\square$

If we ignore the vertex parity, a reflecting cut gives an automorphism of the graph that squares to identity. The $\psi$-graph in figure 3 admits a reflecting cut. It is shown in figure 4.

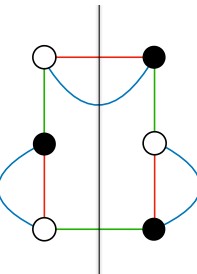

**Figure 4**. The reflecting cut is shown by a straight black line passing through the graph. It is easy to see that the graph is symmetric under the reflection across the reflecting cut, after parity flip.

## 3.1 Constraints from convexity

The invariant $\mathcal{Z}$ constructed from $\psi$'s and $\bar{\psi}$'s can be thought of as a function of the reduced density matrix $\rho_{\bar{A}}$ that is obtained after tracing the party $A$ for some party $A$. The density

matrix $\rho_{\bar{A}}$ is graphically constructed by joining a black and a white vertex only by $A$-edge and keeping all the other edges open as shown in figure 5. In this sense $\rho_{\bar{A}}$ lives on the $A$-edge.

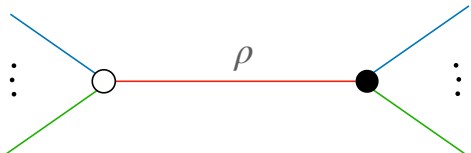

**Figure 5**. Connecting only the $A$-edge (denoted by red color) of a $\psi$-$\bar{\psi}$ pair to get $\rho_{\bar{A}}$.

Alternatively, $\mathcal{Z}$ can also be thought of as a function of $\rho_B$ that lives on $B$-edge and so on.

We plan to use the notion of reflecting cut to investigate the convexity of $\mathcal{Z}$ with respect to $\rho_{\bar{A}}$. Expanding $\mathcal{Z}(\rho_{\bar{A}} + \epsilon \delta \rho_{\bar{A}})$ in $\epsilon$,

$$\mathcal{Z}(\rho_{\bar{A}} + \epsilon \delta \rho_{\bar{A}}) = \mathcal{Z}(\rho_{\bar{A}}) + \epsilon \delta \mathcal{Z} + \frac{1}{2} \epsilon^2 \delta^2 \mathcal{Z} + \mathcal{O}(\epsilon^3). \tag{3.2}$$

Here $\delta \mathcal{Z} = \sum_e \delta_e \mathcal{Z}$ where $\delta_e \mathcal{Z}$ is an invariant obtained from $\mathcal{Z}$ by replacing $\rho_{\bar{A}}$ at $A$-edge $e$ by $\delta \rho_{\bar{A}}$ and similarly, $\delta^2 \mathcal{Z} = \sum_{e',e \neq e'} \delta_e \delta_{e'} \mathcal{Z}$ where $\delta_e \delta_{e'} \mathcal{Z}$ is an invariant obtained from $\mathcal{Z}$ by replacing $\rho_{\bar{A}}$ at $A$-edges $e$ and $e'$ by $\delta \rho_{\bar{A}}$. Graphically, we denote $\delta \rho_{\bar{A}}$ in the same way as $\rho_{\bar{A}}$ but with a dotted line. This is shown in figure 6. $\delta_e \delta_{e'} \mathcal{Z}$ is symmetric in $e, e'$. The

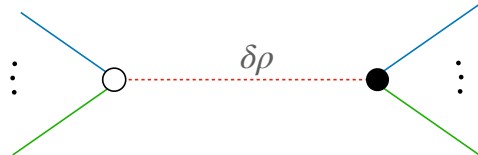

**Figure 6**. The dotted line represents insertion of $\delta \rho_{\bar{A}}$ instead of $\rho_{\bar{A}}$.

convexity of $\mathcal{Z}$ is equivalent to the positivity of $\delta^2 \mathcal{Z}$. We intend to ensure the positivity of $\delta^2 \mathcal{Z} = \sum_{e,e'} \delta_e \delta_{e'} \mathcal{Z}$ by writing it as a sum of norms. For the example of $\psi$-graph in figure 3, we have shown the graphical representation of $\delta \mathcal{Z}$ and $\delta^2 \mathcal{Z}$ below in figure 7.

More concretely, for a general invariant $\mathcal{Z}$, and for a given reflecting cut $k$ of $\mathcal{Z}$, let us denote the open graph on the left side of the cut as $L_k$ and the one on the right side of the cut as $R_k$. Let $R_k$ define the state $|\mathcal{T}^{(k)}\rangle$. Let us also define the state obtained from $|\mathcal{T}^{(k)}\rangle$ by replacing $\rho_{\bar{A}}$ at the $A$-edge $e$ by $\delta \rho$ as $|\mathcal{T}_e^{(k)}\rangle$. Observe that

$$\langle \mathcal{T}^{(k)} | \mathcal{T}_e^{(k)} \rangle = \delta_e \mathcal{Z}, \qquad \langle \mathcal{T}_e^{(k)} | \mathcal{T}^{(k)} \rangle = \delta_{k(e)} \mathcal{Z}, \qquad \langle \mathcal{T}_{e'}^{(k)} | \mathcal{T}_e^{(k)} \rangle = \delta_{k(e')} \delta_e \mathcal{Z}. \tag{3.3}$$

Here $k(e)$ denotes the image of $e$ under the isomorphism defined by the cut $k$. Note that $k(k(e)) = e$.

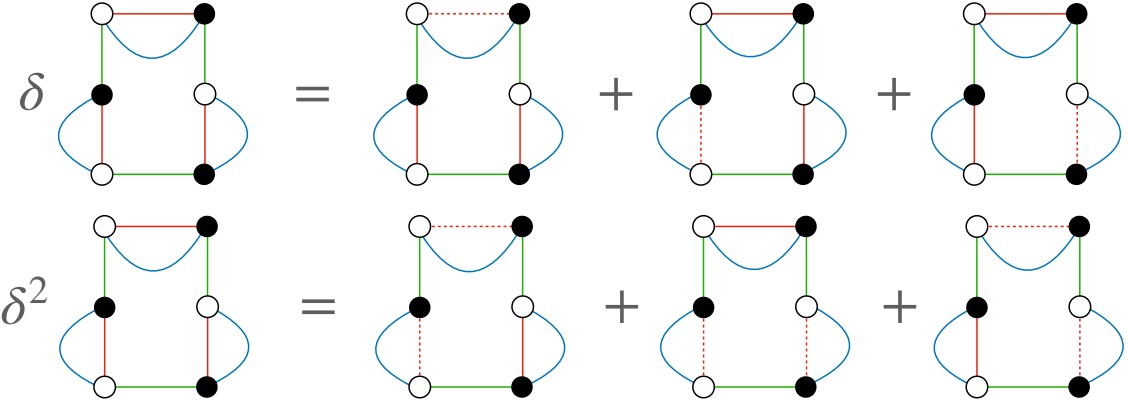

**Figure 7**. The first line denotes $\delta\mathcal{Z}$ for $\mathcal{Z}$ given in figure 3. Each of the $A$-edges has been replaced by $\delta\rho_{\bar{A}}$ and the terms are added up. The second line denotes $\delta^2\mathcal{Z}$ for same $\mathcal{Z}$. A pair of $A$-edges have been replaced by $\delta\rho_{\bar{A}}$ and the terms are added up.

We now consider multiple states of the type $\sum_{e\in R_k} a_e|\mathcal{T}_i^{(k)}\rangle$ for various choices of coefficients and for all reflecting cuts $k$ and add their norms to get the positive quantity,

$$\sum_k \left( \|\sum_{e\in R_k} a_e^{(k)}|\mathcal{T}_e^{(k)}\rangle\|^2 + \|\sum_{e\in R_k} b_e^{(k)}|\mathcal{T}_e^{(k)}\rangle\|^2 + \dots \right) > 0$$

$$\sum_k \sum_{e\in R_k, e'\in L_k} \left( a_e^{(k)} a_{e'}^{(k)*} + b_e^{(k)} b_{e'}^{(k)*} + \dots \right) \delta_e \delta_{e'} \mathcal{Z} > 0$$

$$\sum_k \sum_{e\in R_k, e'\in L_k} \mathcal{M}_{e,e'}^{(k)} \delta_e \delta_{e'} \mathcal{Z} > 0. \tag{3.4}$$

Here $\mathcal{P}_{e,e'}^{(k)} \equiv \mathcal{M}_{e,k(e')}^{(k)}$ where $(e,e') \in R_k$ is a positive definite matrix. This follows from the fact that an arbitrary sum of outer products $aa^\dagger$ can be equivalently written as a positive definite matrix. To obtain convexity of $\mathcal{Z}$, we need to be able to choose positive definite matrices $\mathcal{P}^{(k)}$ such that the left-hand side is $\sum_{e\neq e'} \delta_e \delta_{e'} \mathcal{Z} = \delta^2\mathcal{Z}$. Each term $(e,e')$ with $e \neq e'$ must appear precisely once.

$$\sum_{k\,\text{s.t.}\,e\in R_k, e'\in L_k} \mathcal{M}_{e,e'}^{(k)} = 1. \qquad \forall\, e, e'. \tag{3.5}$$

where the sum $\sum_k$ is carried out over reflecting cuts $k$ which split $e$ and $e'$. If this equation holds for all pairs of $A$-edges then $\mathcal{Z}$ is convex in $\rho_{\bar{A}}$. This equation teaches us that, in particular, for a given pair of vertices $e, e'$, there needs to be at least one reflecting cut separating them so that it has a chance of appearing on the left-hand side sum. This motivates the following definitions.

**Definition 3.2.** *We call a $\psi$-graph $\mathcal{Z}$, A-edge-reflecting if, given a pair of A-edges $(e, e')$, it has the property that there exists a reflecting cut separating them. If $\mathcal{Z}$ is A-edge-reflecting for all parties A then $\mathcal{Z}$ is called edge-reflecting.*

It is clear from the discussion below equation (3.5), that the edge-reflecting condition is a *necessary* condition for $\mathcal{Z}$ to be convex in $\rho_{\bar{A}}$ for all $A$. Because this condition is formulated as a condition on the graph, as we will soon see, it can be effectively studied using techniques of graph theory.

**Definition 3.3.** *If an A-edge-reflecting $\psi$-graph admits solution to equation (3.5) then it is called A-edge-convex. If it is A-edge-convex for all parties then it is called edge-convex.*

It is clear that if a $\psi$-graph $\mathcal{Z}$ is A-edge-convex, it is a convex function of $\rho_{\bar{A}}$. Thanks to theorem 2.1, if $\mathcal{Z}$ is edge-convex then $1 - \mathcal{Z}$ is a PSEM. This combination ensures that the PSEM takes the value 0 on factorized states as required. Unfortunately, unlike the edge-reflecting condition, the condition of edge-convexity is not formulated completely in terms of the graph. In particular, given a graph it is difficult to tell whether it is edge-convex. Nevertheless we will prove a powerful theorem about edge-convex graphs namely theorem 3.2 which we will allow us to construct an infinite family of multi-partite PSEMs.

## 3.2 Some useful lemmas

The discussion in this subsection is somewhat abstract and mathematical. Here we will prove some lemmas which will culminate in theorem 3.1 and theorem 3.2. The reader can skip the proofs during the first read and refer to them when needed.

### 3.2.1 Strengthening convexity

The following lemma will help us strengthen this convexity in $\rho_{\bar{A}}$ following from $A$-edge-convexity.

**Lemma 3.1.** *The automorphism group of A-edge-reflecting graph generated by reflecting cuts acts transitively on A-edges.*

*Proof.* Consider two $A$-edges $e$ and $e'$. It is convenient to collapse all the $A$-edges to vertices so that we get a new graph where each vertex denotes the density matrix $\rho_{\bar{A}}$. Because of connectedness, there exists a path in this graph that connects the edges $e$ and $e'$. Let the

vertices along this path be $e_0, e_1, \ldots, e_n$ with $e_0 = e$ and $e_n = e'$. In the original graph, this is a path that connects the edges $e$ and $e'$ and alternately passes through $A$-edges. Using $A$-edge-reflecting condition, we see that $e_i$ is mapped to $e_{i+1}$ in the original graph with the vertices joined by the common edge getting mapped into each other. The composition of such maps from $e_0$ to $e_n$ gives us the automorphism that maps $e$ to $e'$. $\qquad\square$

Thanks to this transitivity, we have $\delta_e \mathcal{Z} = \delta_{e'} \mathcal{Z}$ for any two $A$-edges $e$ and $e'$. This lets us strengthen the convexity condition on $\mathcal{Z}$ when the solution to (3.5) exists. We now show that the solution to (3.5) not only guarantees the convexity of $\mathcal{Z}$ but in fact of $\mathcal{Z}^{1/n}$ where $n$ is the number of black (or white) vertices in the graph. Considere the state $|\psi\rangle \equiv (\sum_e a_e^{(k)} |\mathcal{T}_i^{(k)}\rangle) \otimes |\mathcal{T}^{(k)}\rangle$ and use the inequality

$$\langle\psi|\psi\rangle \geq \langle\psi|P|\psi\rangle \qquad \text{instead of simply} \qquad \langle\psi|\psi\rangle > 0, \tag{3.6}$$

where $P$ is a unitary operator that acts as a swap on the factors involved in the direct product state $|\psi\rangle$.

$$\mathcal{Z}\Big(\sum_k \sum_{e\in R_k, e'\in L_k} \mathcal{M}_{e,e'}^{(k)} \delta_e \delta_{e'} \mathcal{Z}\Big) \geq \sum_k \sum_{e\in R_k, e'\in L_k} \mathcal{M}_{e,e'}^{(k)} \delta_e \mathcal{Z} \delta_{e'} \mathcal{Z},$$

$$\mathcal{Z}\delta^2\mathcal{Z} \geq |\delta\mathcal{Z}|^2 - \sum_e |\delta_e \mathcal{Z}|^2 = (1 - \frac{1}{n})|\delta\mathcal{Z}|^2. \tag{3.7}$$

For the inequality in the second line, we have used the property of the solution (3.5) and for the equality in the second line, we have used the $A$-edge transitivity. The convexity of $\mathcal{Z}^{1/n}$ now follows because

$$\delta^2(\mathcal{Z}^\alpha) \geq 0 \quad \Leftrightarrow \quad \mathcal{Z}\delta^2\mathcal{Z} - (1-\alpha)|\delta\mathcal{Z}|^2 \geq 0. \tag{3.8}$$

This, along with the fact that $\mathcal{Z} = 1$ for an unentangled state shows that,

**Theorem 3.1.** *If $\mathcal{Z}$ is $A$-edge-convex then $\mathcal{Z}^{1/n}$ is a convex function of $\rho_{\bar{A}}$ where $n$ is the homogeneity of $\mathcal{Z}$ in $\rho_{\bar{A}}$. In particular, if $\mathcal{Z}$ is edge-convex then $\nu = 1 - \mathcal{Z}^{1/n}$ is a* PSEM.

The discussion implies $1 - \mathcal{Z}^\alpha$ for any $\alpha \geq 1/n$ is a PSEM. However, the PSEM $\nu$ with the smallest possible value of $\alpha$, $\alpha = 1/n$, provides the strongest bound on the transition success probability. This is because the PSEMs coming from the higher powers of $\mathcal{Z}$ can be written as $1 - (1-\nu)^\beta$ for $\beta \geq 1$. This function is a concave, monotonic function of $\nu \in [0,1]$ and hence is a composite of $\nu$. Thanks to the lemma A.1, it does not provide stronger bounds than the bounds coming from $\nu$.

**Conversion of multiple copies**

Let us consider the problem of converting $k$ copies of $|\psi\rangle$ to $k$ copies of $|\phi\rangle$. The bound on the transition probability is provided by

$$\frac{\nu(|\psi\rangle^{\otimes k})}{\nu(|\phi\rangle^{\otimes k})} \equiv p_k. \tag{3.9}$$

If $p_1 < 1$, any meaningful bound on this transition probability must have the property that $p_k \geq p_1^k$ because the conversion of $k$ copies of $|\psi\rangle$ to $k$ copies of $|\phi\rangle$ can certainly done copy by copy in $k$ steps. In the same vein, if $k$ is a multiple of $l$, then it must be that $p_k \geq p_l^{k/l}$. This is indeed the property of $\nu$ defined in theorem 3.1. In fact, it obeys stronger inequality, $p_k \geq p_l$. The proof is straightforward and uses the fact $\mathcal{Z}(|\psi\rangle^{\otimes k}) = \mathcal{Z}(|\psi\rangle)^k$.

### 3.2.2   Convexity of cartesian product

Although it may seem unmotivated now, it is useful to define properties analogous to the edge-reflecting condition and the edge-convexity for vertices.

---

**Definition 3.4.** *We call an invariant $\mathcal{Z}$, vertex-reflecting if, given a pair of vertices, it has the property that there exists a reflecting cut separating them.*

---

**Definition 3.5.** *If a vertex-reflecting graph admits solution to equation* (3.5) *with $A$-edges replaced by vertices i.e. to*

$$\sum_{k \,\text{s.t.}\, v \in R_k, v' \in L_k} \mathcal{M}_{v,v'}^{(k)} = 1. \qquad \forall\, v, v'. \tag{3.10}$$

*where the sum is over all reflecting cuts that separate the vertices $v$ and $v'$, then it is called vertex-convex.*

---

The following lemma connects the vertex notions to edge notions.

---

**Lemma 3.2.** *A $\psi$-graph is vertex-reflecting if and only if it is edge-reflecting.*

---

*Proof.*
Edge-reflecting $\Rightarrow$ vertex-reflecting:   If the graph is $K_2$ (complete graph on two vertices) then this is clearly true. Let us assume that the graph is not $K_2$ and hence consists of at

least two types of edges. We would like to show that given any two vertices $i$ and $j$ in the graph we can find a reflecting cut that separates them. There must exist at least one party $A$ such that the $A$-edge incident on vertex $i$ is distinct from the $A$-edge incident on vertex $j$ because if that were not the case, the graph consisting of vertex $i$ and $j$ will be disconnected from the rest. We use the $A$-edge-reflecting condition for this party to find a reflecting cut separating this pair of edges. This cut then also separates the vertices $i$ and $j$. Here we only used that the graph is $A$-edge-reflecting for at least two edge labels $A$.

Vertex-reflecting $\Rightarrow$ edge-reflecting: Given a pair of $A$-edges $e$ and $e'$ for any $A$, we would like to prove the existence of a reflecting cut separating them. Let $e$ join the vertices $u$ and $v$ and $e'$ join the vertices $u'$ and $v'$. These are all distinct as both edges are of the same type. Using vertex-reflecting property, we can find a reflecting cut separating $u$ and $u'$. Let us assume that we can find such a cut that does not cut either $e$ or $e'$. If it does not separate $v$ and $v'$ then the cut graph has a connected path from $u$ to $u'$: $u \to v \to v' \to u'$, a contradiction. So the cut must separate $v$ and $v'$ on the same side of $u$ and $u'$ respectively and hence also separate the edges $e$ and $e'$. Now let us show that we can always find a reflecting cut separating $u$ and $u'$ that cuts neither $e$ nor $e'$. Let us consider a shortest path between $u$ and $u'$. Without loss of generality, let us assume that it does not pass through either $v$ or $v'$ (this can be achieved by swapping the labels $u \leftrightarrow v$ and $u' \leftrightarrow v'$ appropriately). Let us extend this path to $v'$ as $u \to u' \to v'$ and denote it as $p$ and label the vertices along the path as $u_0 = u, u_1, \ldots, u_{|p|-1} = u', u_{|p|} = v'$. The path $p$ is the shortest path between $u$ and $v'$ because the path from $u \to u'$ is the shortest and that $v'$ is $u'$'s neighbor. Consider the reflecting cut $k$ separating $u$ and $u_1$. Clearly, it can not cut the edge $e$ because the image of $u$ under $k$ is $u_1$. It can't also be $v$. Also, due to lemma 3.3, this cut does not cut $p$ anywhere else. So in particular it does not cut $e'$. This completes the proof. $\qquad\square$

If a reflecting cut $k$ is used as a vertex-reflecting cut, we emphasis this by using a subscript as $k_V$.

---

**Lemma 3.3.** *If a reflecting cut separates a pair of vertices $u, v$, then it cuts any shortest path between them precisely once.*

---

*Proof.* Let us number the vertices along the shortest path $p$ as $u_0, u_1, \ldots, u_{|p|}$ such that $u_0 = u$ and $u_{|p|} = v$. Let the reflecting cut separating $u$ and $v$ be $k$. Assuming $k$ cuts the path $p$ more than once, let the first two cuts be right after $u_i$ and $u_j$. The images of $u_i$ and $u_j$ under $k$ are $u_{i+1}$ and $u_{j+1}$. Replacing the segment of the path $u_{i+1} \to u_j$ by its image under $k$, we get a new path between $u$ and $v$ of length $|p| - 2$. This argument is shown graphically in figure 8. This contradicts the assumption that the original path $p$ is the shortest. $\qquad\square$

The above proof also shows us that

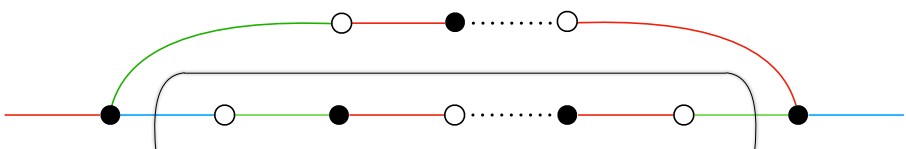

**Figure 8**. The reflecting cut is denoted by a black curve. It cuts the path from $u$ to $v$ in two places. The path above the black curve is the image of the segment in the original path. It is clear that the new path obtained is shorter than the original one by two edges.

**Proposition 3.3.** *The total number of reflecting cuts separating two vertices on an edge-reflecting graph is equal to the length of the shortest path between them.*

*Proof.* Let us number the vertices along some shortest path $p$ between $u$ and $v$ as $u_0, u_1, \ldots, u_{|p|}$ such that $u_0 = u$ and $u_{|p|} = v$. Any reflecting cut separating $u$ and $v$ must cut this path $p$ and moreover it must cut $p$ precisely once. Any edge-reflecting graph is also vertex-reflecting. For a vertex-reflecting graph, for any successive pair of vertices $u_i$ and $u_{i+1}$ along the path $p$, we can find a reflecting cut cutting the edge joining them. Hence the total number of reflecting cuts separating $u$ and $v$ is $|p|$. $\qquad\square$

**Lemma 3.4.** *Any $\psi$-graph that is edge-convex is vertex-convex.*

*Proof.* If the graph is $K_2$, it is vertex-convex. Let us assume that it is not $K_2$. Let us consider all the reflecting cuts $k$ separating $A$-edges. As proved above, those are also the reflecting cuts separating any pair of vertices as long as the pair is not connected by an $A$-edge. For such pairs we define

$$\mathcal{M}_{u,v}^{(k)} = \mathcal{M}_{e_u,e_v}^{(k)} \tag{3.11}$$

where $e_u$ and $e_v$ are the $A$-edges incident on $u$ and $v$ respectively. This is a positive definite matrix. When $u$ and $v$ are connected by $A$-edge, consider a different type of edge, say $B$. The $B$-edge incident on $u$ and $v$ are then distinct. Let $\tilde{k}$ be the reflecting cut separating these $B$-edges. It must cut the $A$-edge joining $u$ and $v$. Because of the reflecting cut property, $u$ and $v$ are images of each other under the reflecting cut $k^*$. This can be done for every pair of $u$ and $v$ that is connected by $A$-edge. Let the associated reflecting cut be $\tilde{k}_{V(u,v)}$. Then for such pairs we take

$$\mathcal{M}_{u',v'}^{(\tilde{k}_{V(u,v)})} = \delta_{u,u'}\delta_{v,v'}. \tag{3.12}$$

This is also a positive definite matrix (a single entry of 1 on the diagonal). This shows that the graph is also vertex-reflecting. □

Note that here also we only used that the graph is $A$-edge-convex with respect to at least two edge labels $A$. The notion of reflecting condition and convexity for vertices helps us prove the theorem,

**Theorem 3.2.** *If the graphs $\mathcal{Z}_1$ and $\mathcal{Z}_2$ are edge-convex then their cartesian product $\mathcal{Z}_1\square\mathcal{Z}_2$ is also edge-convex.*

Let us first recall the definition of the cartesian product of ordinary graphs $G_i$'s. If the vertex and edge set of $G_i$ is denoted as $V_i$ and $E_i$ then the vertex set of their cartesian product $G_1\square G_2$ is the direct product $V_1\otimes V_2$. Two vertices $(v_1,v_2)$ and $(u_1,u_2)$ are joined if either $u_1=v_1$ and $u_2$ and $v_2$ are adjacent to each other in $G_2$ or $u_2=v_2$ and $u_1$ and $v_1$ are adjacent to each other in $G_1$. In other words, the adjacency matrix $A$ of $G_1\square G_2$ is

$$A = A_1\otimes\mathbb{I}+\mathbb{I}\otimes A_2 \tag{3.13}$$

where $A_i$ is the adjacency matrix of $G_i$. The notion of cartesian product is extended straightforwardly for $\psi$-graphs. Because we need $\mathcal{Z}_1\square\mathcal{Z}_2$ to be a $\psi$-graph, it is necessary to assume that the edges of $\mathcal{Z}_1$ and $\mathcal{Z}_2$ do not have a common label. Then the vertex parity of $(v_1,v_2)$ is the sum of parities of $v_1$ and $v_2$. As for the edge-labels, we simply used the formula (3.13) separately for every label $A$ in defining the adjacency matrix of $\mathcal{Z}_1\square\mathcal{Z}_2$. It is this extension of cartesian product that we will always use and denote by the symbol $\square$. See figure 9 for an example. We are now ready to prove the theorem 3.2.

*Proof.* First let us show that if $\mathcal{Z}_1$ and $\mathcal{Z}_2$ are edge-reflecting then $\mathcal{Z}_1\square\mathcal{Z}_2$ is also edge-reflecting. Without loss of generality let us assume that $\mathcal{Z}_1$ is $A$-edge-reflecting and that $\mathcal{Z}_2$ does not have any $A$-edge. The $A$-edges in the cartesian product will be labeled by the pair $(e,u)$ and $(f,v)$ where $e$ and $f$ are $A$-edges of $\mathcal{Z}_1$ and $u,v$ are vertices of $\mathcal{Z}_2$. If $e$ and $f$ are distinct edges of $\mathcal{Z}_1$, we can simply use the reflecting cut that separates them in $\mathcal{Z}_1$ and take its cartesian product with $\mathcal{Z}_2$ to produce the required reflecting cut in $\mathcal{Z}_1\square\mathcal{Z}_2$. If $e=f$ and $u\neq v$, we find a reflecting cut in $\mathcal{Z}_2$ separating $u$ and $v$ using its vertex-reflecting property. Taking its cartesian product with $\mathcal{Z}_1$ produces the required reflecting cut.

Now we construct the solution to (3.5) for $\mathcal{Z}_1\square\mathcal{Z}_2$ using a similar strategy. using the solution to the same equation for $\mathcal{Z}_1$ and $\mathcal{Z}_2$ separately. We would like to show that the pair of $A$-edges, $(e,u)$ and $(f,v)$ appears with weight 1. Denoting the reflecting cuts separating $e$ and $f$ in $\mathcal{Z}_1$ as $k$,

$$\mathcal{M}^{(\tilde{k})}_{(e,u),(f,v)} = \mathcal{M}^{(k)}_{e,f} \qquad \forall u,v. \tag{3.14}$$

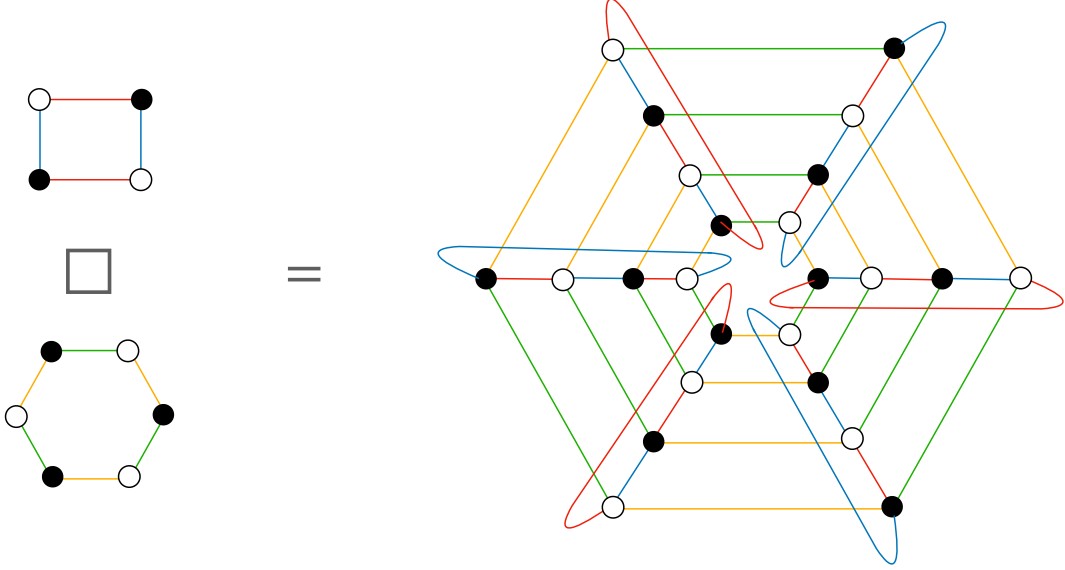

**Figure 9**. Example of the colored cartesian product.

Here $\tilde{k} = k \square \mathcal{Z}_2$ is a reflecting cut of $\mathcal{Z}_1 \square \mathcal{Z}_2$. As $\mathcal{P}_{e,f}$ is positive definite, $\mathcal{P}_{(e,u),(f,v)}$ is also positive definite. This will create all the pairs of $A$-edges with weight 1 except for the ones where $e = f$ and $u \neq v$. To produce these remaining terms with weight 1, we use the vertex-convexity of graph $\mathcal{Z}_2$. Let us denote the reflecting cuts separating $u$ and $v$ of $\mathcal{Z}_2$ as $k_V$.

$$\mathcal{M}^{(\tilde{k})}_{(e,u),(f,v)} = \mathcal{M}^{(k_V)}_{u,v}\, \delta_{e,f}. \tag{3.15}$$

Here $\tilde{k} = \mathcal{Z}_1 \square k_V$ is a reflecting cut of $\mathcal{Z}_1 \square \mathcal{Z}_2$. For an edge-convex graph, the matrix $\mathcal{M}^{(k_V)}_{u,v}$ is constructed in lemma 3.4. $\qquad\square$

## 3.3 A concrete example

Let us put this theory on a firmer footing by considering some examples of edge-convex graphs. Consider the $\psi$-graph that is 1-skeleton (graph made by the vertices and edges) of a q-dimensional hypercube. The vertex parity is the parity of the binary coordinate of the vertex and all edges that are "parallel" have the same labels. Let us denote this graph as $\mathcal{E}^{(\mathsf{q})}$. It has q-colors, so it is an invariant of a q-partite state. Moreover, it has the property that it is symmetric under permutations of all the parties. It is clear that $\mathcal{E}^{(\mathsf{q})} = \mathcal{E}^{(\mathsf{r})} \square \mathcal{E}^{(\mathsf{q}-\mathsf{r})}$ for any $\mathsf{r} < \mathsf{q}$. It is also clear that $\mathcal{E}^{(1)} = K_2$ is edge-convex. This, along with theorems 3.2 and 3.1, proves

**Corollary 3.1.** $1 - (\mathcal{E}^{(\mathsf{q})})^{2^{1-\mathsf{q}}}$ *is a* PSEM.

The exponent $2^{1-\mathsf{q}}$ is simply the inverse of the number of $\psi$'s in $\mathcal{E}^{(\mathsf{q})}$. Corollary 3.1 is one of the main results of the paper. Let us explicitly demonstrate the edge-convexity of $\mathcal{E}^{(\mathsf{q})}$ for $\mathsf{q} = 2$, $\mathsf{q} = 3$ and $\mathsf{q} = 4$ by following the construction given in lemma 3.2.

First, $\mathcal{E}^{(1)} = K_2$ has two vertices and a single edge, say of type $A$, connecting them. As there is only a single edge, the graph is automatically edge-reflecting as there is no pair of edges that needs separating. It has two vertices and is vertex-reflecting. The associated reflecting cut $k_V$ simply cuts the edge. See figure 10 So $\mathcal{P}^{(k_V)} = (1)$.

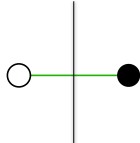

**Figure 10**. Graph of $\mathcal{E}^{(1)} = K_2$. The black line denotes the reflecting cut $k_V$

The $\psi$-graph $\mathcal{E}^{(2)}$ is a square as shown in figure 11. We have also shown the reflecting cut $k$ separating $A$-edges (denoted by red color). The associated positive matrix is $\mathcal{P}^{(k)} = (1)$.

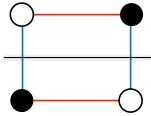

**Figure 11**. Graph of $\mathcal{E}^{(2)}$. The black line denotes the reflecting cut $k$

The $\psi$-graph $\mathcal{E}^{(3)}$ is a cube as shown in figure 12, along with the two reflecting cuts $k_1$ and $k_2$ separating $A$-edges. Thinking of $\mathcal{E}^{(3)} = \mathcal{E}^{(2)} \Box \mathcal{E}^{(1)}$, and $k_1 = k \Box \mathcal{E}^{(1)}$ where $k$ is

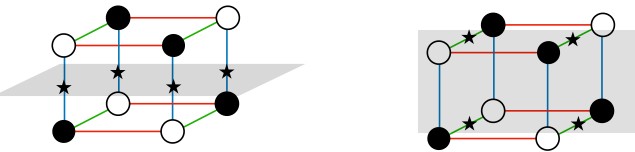

**Figure 12**. Graph of $\mathcal{E}^{(2)}$ and the two reflecting cuts are suggestively denoted by the gray sheets. The first one is $k_1 = k \Box \mathcal{E}^{(1)}$ and the second one is $k_2 = \mathcal{E}^{(2)} \Box k_V$.

the reflecting cut of $\mathcal{E}^{(2)}$, and $k_2 = \mathcal{E}^{(2)} \square k_V$ where $k_V$ is the reflecting cut associated to vertex-convexity of $\mathcal{E}^{(1)}$,

$$\mathcal{P}^{(k_1)} = \begin{pmatrix} 1 & 1 \\ 1 & 1 \end{pmatrix} \qquad \mathcal{P}^{(k_2)} = \begin{pmatrix} 1 & 0 \\ 0 & 1 \end{pmatrix}. \tag{3.16}$$

This is not a unique solution for $\mathcal{P}^{(k_1)}$ and $\mathcal{P}^{(k_2)}$. Following is a symmetric solution.

$$\mathcal{P}^{(k_1)} = \mathcal{P}^{(k_2)} = \begin{pmatrix} 1 & \frac{1}{2} \\ \frac{1}{2} & 1 \end{pmatrix}. \tag{3.17}$$

Now consider $\mathcal{E}^{(4)}$ and think of it as $\mathcal{E}^{(3)} \square \mathcal{E}^{(1)}$. It has three reflecting cuts $k_1'$, $k_2'$ and $k_3'$. Let us think of them as $k_1 \square \mathcal{E}^{(1)}$, $k_2 \square \mathcal{E}^{(2)}$ and $\mathcal{E}^{(3)} \square k_V$ respectively, where $k_1, k_2$ are reflecting cuts of $\mathcal{E}^{(3)}$ and $k_V$ is the reflecting cut for vertex-convexity of $\mathcal{E}^{(1)}$. Then,

$$\mathcal{P}^{(k_1')} = \begin{pmatrix} 1 & 1 & 1 & 1 \\ 1 & 1 & 1 & 1 \\ 1 & 1 & 1 & 1 \\ 1 & 1 & 1 & 1 \end{pmatrix}. \qquad \mathcal{P}^{(k_2')} = \begin{pmatrix} 1 & 0 & 1 & 0 \\ 0 & 1 & 0 & 1 \\ 1 & 0 & 1 & 0 \\ 0 & 1 & 0 & 1 \end{pmatrix}. \qquad \mathcal{P}^{(k_3')} = \begin{pmatrix} 1 & 0 & 0 & 0 \\ 0 & 1 & 0 & 0 \\ 0 & 0 & 1 & 0 \\ 0 & 0 & 0 & 1 \end{pmatrix}. \tag{3.18}$$

Again, this is not a unique solution and one can find a symmetric solution,

$$\mathcal{P}^{(k_1')} = \mathcal{P}^{(k_2')} = \mathcal{P}^{(k_3')} = \begin{pmatrix} 1 & \frac{1}{2} & \frac{1}{2} & \frac{1}{3} \\ \frac{1}{2} & 1 & \frac{1}{3} & \frac{1}{2} \\ \frac{1}{2} & \frac{1}{3} & 1 & \frac{1}{2} \\ \frac{1}{3} & \frac{1}{2} & \frac{1}{2} & 1 \end{pmatrix}. \tag{3.19}$$

### 3.4   General solution to edge-reflecting condition

We would like to find a general solution to the edge-convexity condition to produce all PSEMs that are polynomial in $\psi$. However, solving (3.5) in full generality is hard. Instead we will find a large class of solutions to edge-reflecting condition for a subset of $\psi$-graphs called simple $\psi$-graphs. This condition is a *necessary* condition for edge-convexity. In the next subsection, we will also show that a particular family $\mathcal{C}_n$ of the solution to the edge-reflecting condition satisfies (3.5) and hence is edge-convex. A vast class of PSEMs can then be constructed by taking Cartesian products of $\mathcal{C}_n$'s.

---

**Definition 3.6.** *A graph is called simple if it has at most one edge between a pair of vertices.*

---

In what follows, we will only consider the characterization of simple $\psi$-graphs. We need to introduce a slightly coarser structure first.

**Definition 3.7.** *For a set of* q *colors, if every vertex of the graph has precisely* q*-edges, one of each color then such a graphs is called a color-regular graph.*

It is clear that any $\psi$-graph is a color-regular graph and moreover, any bi-partite color-regular graph is a $\psi$-graph. In fact, the $\psi$-graph obtained from a color-regular graph is unique (up to the overall vertex parity flip). In the rest of the section, we will forget the vertex parity of $\psi$-graphs and treat them as only color-regular graphs. After understanding their structure, we will specialize to bi-partite color regular graph. We first extend the notion of reflecting cut to color-regular graphs by ignoring the vertex parity flip condition. We will now solve edge-reflecting conditions for simple color-regular graphs. Just like for $\psi$-graph, the edge-reflecting condition implies vertex-reflecting condition on color-regular graphs. We have,

**Lemma 3.5.** *The group of automorphisms of a vertex-reflecting color-regular graph, generated by the reflecting cuts, acts transitively on vertices.*

*Proof.* The proof is similar to the proof of edge-transitivity in lemma 3.1. $\qquad\square$

Thanks to [21], vertex transitivity turns out to be a strong constraint on color-regular graphs. In fact, this condition characterizes a color-regular graph completely as a Cayley graph of some group G with some distinguished generators S. We will not go into the proof of this, as it will take us into automata theory[1], but rather only state the result of [21] precisely.

**Definition 3.8.** *A directed edge-colored graph $G$ is called* sdrv *graph if it satisfies the following conditions.*

- *It has at most one directed edge from one vertex to the other (called* simple *in [21]).*

- *It has at most one edge of a given label and given orientation incident on any vertex (called* deterministic *in [21]).*

- *It has at least one vertex from which every other vertex can be reached following a directed path (called* rooted *in [21]).*

- *It is vertex-transitive i.e. a subgroup of the automorphism group of the graph acts transitively on vertices.*

---

[1]For our purposes, automata theory is study of groups using "words" formed by "letters" which are generators of the group.

It might seem that an `sdrv` graph is a new structure unrelated to what we want to consider, namely simple color-regular graphs but that is not see. If we replace all the (undirected) edges of the simple color-regular graph by a pair of edges directed both ways then we get an `sdrv` graph. Of course, a general `sdrv` graph is not related to any color-regular graph simple or not. It is proved in [21] that,

**Theorem 3.3** (Caucal [21]). *An `sdrv` graph is a Cayley graph.*

A closely related result had been proved by Sabidussi [22] which assumes that the graph action is fixed-point-free. Caucal's result can be thought of as replacing the condition of fixed-point-free-ness with conditions involving structural properties of the graph i.e. graph being simple, deterministic and rooted. In the absence of any obvious fixed-point-free condition, it is Caucal's result that is more useful to us. Let us recall the definition of a Cayley graph. A Cayley graph is a graph associated with a group $G$ *and* a set $S$ of its generators. It has group elements for its vertices and an edge starts from $g_1$ and goes to $g_2$ if and only if $g_2 = a \cdot g_1$ for some $a \in S$. This edge is labeled by the element $a$.

The pair $(G, S)$ associated with an `sdrv` graph, thought of as a Cayley graph is also constructed in [21] using the tools of automata theory. This pair will be very useful for us in characterizing simple color-regular graphs. We will only sketch the construction here. As the graph $G$ is vertex-transitive, without loss of generality, consider any vertex $v$ of the graph and look at the edges leaving it. Because the graph is deterministic, they all have distinct labels. The set of these labels forms the set of group generators $S$. The words associated with all the directed paths that start and end at $v$ are taken to be the only relations on these generators. This gives an explicit presentation of the group $G$ in terms of its generators.

Using this construction, the pair $(G, S)$ associated to the `sdrv` graph arising from a simple color-regular graph can be readily found. Each vertex has edges with $q$ edge-labels $A_1, \ldots, A_q$. Each of them is a generator. And because each edge is bidirectional, all the generators obey $A_a^2 = 1$ for all $q = 1, \ldots, q$. Recall that a $\psi$-graph is a bi-partite color-regular graphs. A graph is bi-partite if and only if it does not have any odd cycles. So all other relations on the generators must have an even number of $A$'s. In particular, for a pair of labels $(A_a, A_b)$ there must exist a finite directed cycle that alternates i.e. takes the form $A_a A_b A_a A_b \ldots$. As such a cycle is even, the associated relation must take the form $(A_a A_b)^{m_{ab}} = 1$. There could be even more relations consisting of an even number of $A$s. The groups with generators $A_a$ obeying $A_a^2 = 1$ and $(A_a A_b)^{m_{ab}} = 1$ are known as Coxeter groups. This shows that

**Theorem 3.4.** *A simple edge-reflecting $\psi$-graph is the Cayley graph of a quotient group of a Coxeter group where the quotient is by relations consisting of even number of generators.*

How about the other way around? Which quotient groups of Coxeter groups are edge-reflecting $\psi$-graphs? We find this difficult to answer in general. However, we show that

**Theorem 3.5.** *The Cayley graph of a Coxeter group is an edge-reflecting $\psi$-graph.*

*Proof.* To see this we consider the Weyl presentation of the Coxeter group. For each of the q generators $A_{\mathsf{a}}$, we place a co-dimension-1 mirror passing through the origin in $\mathbb{R}^{\mathsf{q}}$. The angle between the two mirror faces associated with $A_{\mathsf{a}}$ and $A_{\mathsf{b}}$ is taken to be $\pi/m_{\mathsf{ab}}$. This fixes all the mirror placements. We place a vertex in the chamber enclosed by all the mirrors. Its reflections through all the mirrors generate the vertex set of our graph. Two vertices that are reflections of each other through the mirror corresponding to $A_{\mathsf{a}}$, we join them by an $A_{\mathsf{a}}$-edge. This graph is bi-partite as the orientation of a chamber flips through each reflection. In this way, we have realized a $\psi$-graph with Weyl reflections. The Coxeter group acts freely and transitively on the vertices as expected. Each of the mirrors and their images gives a reflecting cut of the graph. If this graph were not vertex-reflecting then there would exist a pair of vertices that is in the same chamber. This is certainly not true by construction. So the $\psi$-graph thus constructed is vertex-reflecting and hence by lemma 3.2, it is edge-reflecting. $\square$

Interestingly, the edge-reflecting property of the Cartesian product of $\psi$-graphs that are Cayley graphs of Coxeter groups is automatic. If $\mathcal{Z}_{G_i}$ is a Cayley graph of the Coxeter group $G_i$ thought of as $\psi$-graph. Then their Cartesian product is

$$\mathcal{Z}_{G_1} \square \mathcal{Z}_{G_2} = \mathcal{Z}_{G_1 \square G_2} \tag{3.20}$$

where $G_1 \square G_2$ is a Coxeter group whose Coxeter diagram consists of a disjoint union of Coxeter diagrams of $G_1$ and $G_2$.

Theorems 3.4 and 3.5 almost completely characterize edge-reflecting $\psi$-graphs. It would be interesting to study edge-reflecting properties of the quotient groups of Coxeter groups in the future.

### 3.5 Edge-convexity

In this section, we will consider the edge-reflecting graph $\mathcal{C}_n$, shown in figure 13 and show that it is edge-convex. It is a cyclic graph with $n$ $\psi$s and $n$ $\bar{\psi}$s with edges alternating between types $A$ and $B$. The edges $A$ and $B$ are shown in the figure with the colors red and blue respectively. Thought of as a local unitary invariant for two parties $A$ and $B$, $\mathcal{C}_n$ is nothing but $\mathrm{Tr}\rho_A^n$ or equivalently $\mathrm{Tr}\rho_B^n$.

We have $\mathcal{C}_1 = \mathcal{E}^{(1)} = K_2$ and $\mathcal{C}_2 = \mathcal{E}^{(2)}$. We have already discussed the edge-convexity of these $\psi$-graphs. For $n \geq 3$, we have new edge-reflecting graphs. In general, $\mathcal{C}_n$ has $n$ reflecting cuts, as shown in figure 13. For every reflecting cut, we take the positive matrix to be the identity matrix. This is a positive definite matrix. Given a pair of $A$-type edges, there is

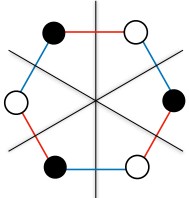

**Figure 13**. Graphical presentation of $\mathcal{C}_3$. It is edge-reflecting. We have denoted the three reflecting cuts with black lines. The graph for $\mathcal{C}_n$ is cyclic with $n$ $\psi$s and $n$ $\bar{\psi}$s with edges alternating between types $A$ and $B$. It has $n$ reflecting cuts.

precisely one cut such that these edges are images of each other under it. This ensures that $\mathcal{P}^{(k)} = \mathbb{I}$ for all $k$ indeed solve the equation (3.5). This proves that $\mathcal{C}_n$ are edge-convex and so we have,

**Lemma 3.6.** $1 - (\mathcal{C}_n)^{1/n}$ *is a* PSEM.

Taking cartesian product, this implies

**Theorem 3.6.** $1 - (\mathcal{C}_{n_1} \square \ldots \square \mathcal{C}_{n_k})^{1/(2^{k-1} n_1 \ldots n_k)}$ *is a* PSEM.

Every $\mathcal{C}_n$ in the Cartesian product contributes 2 edge-colors for $n > 1$. If $n = 1$, then it contributes 1 color[2]. The cartesian product, $\mathcal{C}_3 \square \mathcal{C}_1$, shown in figure 14 yields a tri-partite PSEM. When we take all $n_i$ to be 1, we get the PSEM $1 - (\mathcal{E}^\mathsf{q})^{2^{1-\mathsf{q}}}$ discussed in corollary 3.1. Note that the quantity $\mathcal{C}_n \square \mathcal{C}_1$ is analytic in $n$ because it is the $n$-th power of a certain matrix.

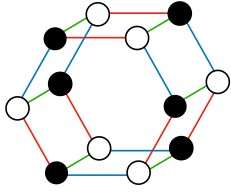

**Figure 14**. Graphical presentation of $\mathcal{C}_3 \square \mathcal{C}_1$.

---

[2]We can also think of it as having two edges between two vertices with different colors. These edges are combined into one by thinking of the two parties as a single party

Interestingly, its analytic continuation to $n = 1/2$ gives what is known as the computable cross-norm (CCNR), first introduced to address the problem of separability in [23, 24]. The CCNR has also been discussed in the context of $2d$ conformal field theory in [25] and in the context of holography in [26].

The graph $\mathcal{C}_n$, thought of as a directed sdrv graph, is the Cayley graph of the simplest Coxeter group $\mathbb{Z}_n$. In theorem 3.5, we prove that Cayley graph of any Coxeter group is edge-reflecting and as discussed near equation (3.20), the graphs obtained by taking cartesian products of $\mathcal{C}_n$ is also the Cayley graph of a Coxeter group. This leads us to conjecture,

> **Conjecture 3.1.** *The Cayley graph of a Coxeter group is an edge-convex $\psi$-graph.*

It would be interesting to explore this conjecture further. If true, it would vastly enrich the set of PSEMs which we can use to bound transition probabilities.

Incidentally, the multi-partite invariants corresponding to the Cayley graphs of a Coxeter groups also appear as a solution to a completely different problem. In [27], authors investigate multi-partite invariants for holographic conformal field theories with bulk duals that preserve replica symmetry. They observe that invariants corresponding to the Cayley graph of a Coxeter group have this property. It would be interesting to explore this connection further and to use the conjectured locc monotonicity property of such invariants to learn new aspects of holographic states.

## 4 Example of a bound

Using the PSEMs constructed in theorem 3.6, we will give bounds on transition success probability for tri-partite states $|\psi\rangle \rightarrow |\phi\rangle$. In particular, we will take $|\psi\rangle$ to be a GHZ-type state

$$|\psi\rangle = \frac{1}{\sqrt{5}}(2|000\rangle + |111\rangle) \tag{4.1}$$

Let us denote the three qubits as $A, B, C$. As discussed near equation (2.10), in order to have a non-zero transition probability to $|\phi\rangle$, $|\phi\rangle$ needs to be related to $|\psi\rangle$ by an slocc transformation. We take $|\phi\rangle$ to be,

$$|\phi\rangle = M_1 \otimes M_2 \otimes M_3 |\psi\rangle|_{\text{normalized}}. \tag{4.2}$$

For convenience, we will take $M_1, M_2$ and $M_3$ to be identical and equal to $\exp(\alpha K)$ for some real $\alpha$. We choose $K$ arbitrarily to be,

$$K = \begin{pmatrix} 1 & 1 \\ -2 & -1 \end{pmatrix}. \tag{4.3}$$

This gives us a family of states $|\phi_\alpha\rangle$, all of which are related to $|\psi\rangle$ by locc. Because all $M_i$ are chosen to be identical, just like $|\psi\rangle$, $|\phi\rangle$ is also symmetric under permutation of parties.

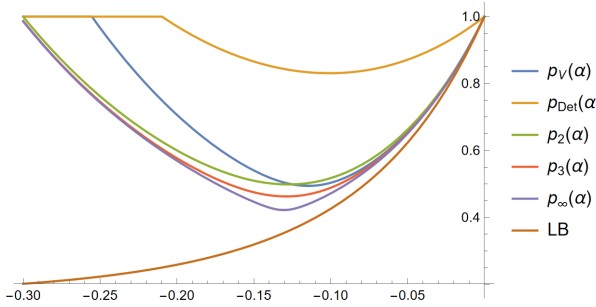

**Figure 15**. Plot of $p_V(\alpha)$ and $p_n(\alpha)$ for $n = 2, 3, 4$. We have also plotted the lower bound coming from equation (2.11).

We first compute the optimal transition probability $p_V$ for $|\psi\rangle \to |\phi_\alpha\rangle$, by thinking of them as bi-partite states under the bi-partition, say $A$ and $BC$. Due to the symmetry of the state, all the bi-partitions are equivalent.

$$p_V(\alpha) = \min_k \frac{\tilde{\nu}_k^V(|\psi\rangle)}{\tilde{\nu}_k^V(|\phi_\alpha\rangle)}. \tag{4.4}$$

This is the optimal transition probability of $|\psi\rangle \to |\phi_\alpha\rangle$ if qubits $B$ and $C$ can be brought together and a joint operation can be performed on them. In the absence of such a joint operation, we expect that this transition probability to be smaller. We use the tri-partite monotones $\nu_n \equiv 1 - (\mathcal{C}_1 \Box \mathcal{C}_n)^{1/2n}$, that we constructed in theorem 3.6, to put upper bound $p_n$ on this "tri-partite" probability.

$$p_n(\alpha) = \frac{\nu_n(|\psi\rangle)}{\nu_n(|\phi_\alpha\rangle)}. \tag{4.5}$$

We have plotted $p_V(\alpha)$ as well as $p_n(\alpha), n = 2, 3, \infty$ for a range of $\alpha$ in figure 4. This figure effectively illustrates the central point of the paper: we see that for a range of $\alpha$'s, the transition probability for $|\psi\rangle \to |\phi_\alpha\rangle$ that is achievable by truly local processes is smaller than the probability if the processes were non-local.

We have also plotted the bound on the transition probability obtained from the PSEM $\nu_{\text{Det}}$ of equation (2.13),

$$p_{\text{Det}}(\alpha) = \frac{\nu_{\text{Det}}(|\psi\rangle)}{\nu_{\text{Det}}(|\phi_\alpha\rangle)}. \tag{4.6}$$

We see that the bounds obtained from the PSEMs constructed here are more stringent than $p_{\text{Det}}(\alpha)$ bound for the given range of $\alpha$. As a sanity check we have also plotted the lower bound on the transition probability $|\psi\rangle \to |\phi_\alpha\rangle$ computed in equation (2.11). As expected, it is below all the upper bounds discussed here.

It would be interesting to investigate if the multi-partite entanglement monotones constructed here are primitive as defined in Definition 1.4. Another important question, along these lines, is to find the minimal complete set of multi-partite entanglement monotones and show that it is minimal by explicitly constructing the optimal transformation protocol.

## Acknowledgements

We would like to thank Jonathan Harper, Vineeth Krishna, Gautam Mandal, Shiraz Minwalla, Onkar Parrikar, Trakshu Sharma, Piyush Shrivastava, Sandip Trivedi for interesting discussions. This work is supported by the Infosys Endowment for the study of the Quantum Structure of Spacetime and by the SERB Ramanujan fellowship. We acknowledge the support of the Department of Atomic Energy, Government of India, under Project Identification No. RTI 4002. HK would like to thank KVPY DST fellowship for partially supporting his work. Finally, we acknowledge our debt to the people of India for their steady support to the study of the basic sciences.

## A    Bound from composites is weaker

**Theorem A.1.** *If $G(x_1, \ldots, x_k)$ is a positive, monotonic and concave function of its arguments $x_i \geq 0$ in some domain then in that domain,*

$$\frac{G(x_1, \ldots, x_k)}{G(x'_1, \ldots, x'_k)} \geq \min\left(\frac{x_1}{x'_1}, \ldots, \frac{x_k}{x'_k}, 1\right). \tag{A.1}$$

*Proof.* We will prove this inequality using induction in $k$. Let first us assume that we have proved it for $k = 1$. Consider the case $x_k < x'_k$, then

$$\frac{G(x_1, \ldots, x_k)}{G(x'_1, \ldots, x'_k)} = \frac{G(\alpha_1\, x_k, \ldots, \alpha_{k-1}\, x_k, x_k)}{G(\alpha_1 x'_k, \ldots, \alpha_{k-1}\, x'_k, x'_k)} \times \frac{G(\alpha_1 x'_k, \ldots, \alpha_{k-1}\, x'_k, x'_k)}{G(\beta_1\, x'_k, \ldots, \beta_{k-1}\, x'_k, x'_k)} \tag{A.2}$$

Here we have defined $\alpha_i = x_i/x_k$ and $\beta_i = x'_i/x'_k$ for $i = 1, \ldots, k-1$. Let the first ratio on the right hand side be $A$ and the second be $B$. Define $f(x_k) = G(\alpha_1\, x_k, \ldots, \alpha_{k-1}\, x_k, x_k)$. This is a monotonic and concave function of $x_k$. This shows

$$A \geq \min\left(\frac{x_k}{x'_k}, 1\right) = \frac{x_k}{x'_k}. \tag{A.3}$$

Now define $\tilde{f}(\alpha_1, \ldots, \alpha_{k-1}) = G(\alpha_1 x'_k, \ldots, \alpha_{k-1}\, x'_k, x'_k)$. This is a monotonic and concave function of its $k - 1$ arguments. Using induction,

$$B \geq \min\left(\frac{\alpha_1}{\beta_1}, \ldots, \frac{\alpha_{k-1}}{\beta_{k-1}}, 1\right). \tag{A.4}$$

The inequalities (A.3) and (A.4) imply

$$A \cdot B \geq \min\left(\frac{x_1}{x'_1}, \ldots, \frac{x_k}{x'_k}, 1\right). \tag{A.5}$$

In the other case $x_k \geq x'_k$,

$$\frac{G(x_1,\ldots,x_k)}{G(x'_1,\ldots,x'_k)} \geq \frac{G(x_1,\ldots,x_{k-1},x'_k)}{G(x'_1,\ldots,x'_{k-1},x'_k)} \geq \min\left(\frac{x_1}{x'_1},\ldots,\frac{x_{k-1}}{x'_{k-1}},1\right) \geq \min\left(\frac{x_1}{x'_1},\ldots,\frac{x_k}{x'_k},1\right).$$
(A.6)

Here, in the first inequality, we have used the monotonicity property of $G$ i.e. $G(x_1,\ldots,x_{k-1},x'_k) \geq G(x_1,\ldots,x_{k-1},x_k)$. In the second inequality, we have used the inductive assumption for the function $f'(x_1,\ldots,x_{k-1}) = G(x_1,\ldots,x_{k-1},x'_k)$ of $k-1$ arguments.

Now, let us prove the inequality (A.1) for $k = 1$. For $x_1 \geq x'_1$,

$$\frac{G(x_1)}{G(x'_1)} \geq 1,$$
(A.7)

due to monotonicity of $G$. We only need to consider the case $x_1 < x'_1$.

$$\frac{G(x_1)}{G(x'_1)} - \frac{x_1}{x'_1} = \frac{x'_1 G(x_1) - x_1 G(x'_1)}{x'_1 G(x'_1)}.$$
(A.8)

Now we will show that the numerator $n(x,y) = yG(x) - xG(y)$ of the right hand side is non-negative for $x < y$.

$$\partial_y n(x,y) = G(x) - xG'(y)$$

$$\partial_x \partial_y n(x,y) = G'(x) - G'(y) = -\int_x^y d\alpha\, G''(\alpha) \geq 0.$$
(A.9)

This shows that the function $\partial_y n(x,y)$ is a monotonic function of $x$. So

$$G(x) - xG'(y) \geq G(0) - 0G'(y) \geq 0.$$
(A.10)

This shows that the function $n(x,y)$ is a monotonic function of $y$. Hence $n(x,y) \geq n(x,x) = 0$. This completes the proof. $\qquad\square$

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
