# Peer review of "Multi-partite entanglement monotones"

_SciPost Physics_

## Round 3 · Referee Report · Anonymous (Referee 1) · 2025-11-27

Report

In the manuscript "Multi-partite entanglement monotones" the authors consider pure state entanglement measures given by local unitary (LU) invariant homogeneous polynomials. After recalling the seminal results on measuring entanglement, they describe LU-invariant polynomials by graphs encoding the index contractions. The main result of the manuscript is the notion of edge-convexity of the graph, ensuring the entanglement monotonicity of the LU-invariant given by that graph. The derivations seem to be correct, although the sloppy way of presentation hinders the understanding and checking a lot. Edge-convexity is, in fact, just the reformulation of the convexity of the LU-invariants in graph language.

Although the topic is rather important and interesting, in my opinion the results do not make a contribution to scholarship significant enough to reach the level of publication in prominent journals like SciPost.

Even in the case of the submission in another journal, I would suggest a thorough revision of the presentation. In general, a liberal style in writing works if the authors have the trust of the reader about that the steps are correct, even if not written down precisely. This trust usually exists in the beginning of the reading, but the sloppy way of writing undermines it. I just list some points. (Please do not consider this list complete, I simply do not have more time to write down all the criticism.)

In Definition I.2, - The third property should be called "faithfulness", it has nothing to do with normalization. Normalization would be that it takes the value 1 for a Bell-pair. - In the first property, it should be written that this has to hold for all subsystems, not just for A.

It is also confusing that A, B, C sometimes denote labels of concrete elementary subsystems (for example, in the second paragraph, or in the sentence following Definition II.1), sometimes variables of labels of concrete elementary subsystems (for example in Equation (19)); and sometimes the labels of elementary subsystems are natural numbers ranging from 1 to q (for example in Equations (4) or (19)). (Using the quite distinctive font for the q number of elementary subsystems is also not a good style. Using such font for the abbreviations is that too. locc and slocc are abbreviations too, so should be capitalized as LOCC, SLOCC.) It is even worse that A sometimes denotes an operator (for example directly after Equation (19)).

"As remarked earlier, the condition of convexity is not physically motivated and is optional. Indeed logarithmic negativity is an important monotone that is not convex [4]." I would not say this, convexity is important and highly motivated physically. Logarithmic negativity is simply not a good measure of entanglement. Not even faithful.

In Definition II.1: - \nu(\ket{\psi}) appears twice, - \ket should be used in the last equality of (12) - linear operators acting on the Hilbert space do not (cannot) preserve the trace, the conjugation by those do.

In Definition II.2: - \mu is in the minimization, but \nu in the formula minimized.

The use of the name "multi-invariant" sounds pretty naive. The algebra of LU-invariant homogeneous polynomials is a relatively well-understood topic, see - Michael W. Hero, Jeb F. Willenbring, Stable Hilbert series as related to the measurement of quantum entanglement, Discrete Mathematics 309, 6508 - Michael W. Hero, Jeb F. Willenbring, and Lauren Kelly Williams, The measurement of quantum entanglement and enumeration of graph coverings, in Representation Theory and Mathematical Physics (AMS book, DOI:10.1090/conm/557). - Peter Vrana, On the algebra of local unitary invariants of pure and mixed quantum states, Journal of Physics A: Mathematical and Theoretical, 44 225304 - Péter Vrana, Local unitary invariants for multipartite quantum systems, Journal of Physics A: Mathematical and Theoretical, 44 115302 - Szilárd Szalay, All degree 6 local unitary invariants of k qudits, Journal of Physics A: Mathematical and Theoretical, 45 065302 - Péter Vrana, An algebraically independent generating set of the algebra of local unitary invariants, arXiv:1102.2861 [quant-ph] Some of these also contains the formulation of LU-invariant homogeneous polynomials by graphs.

The structure of the manuscript is also hard to follow, the proper way of using LaTeX's sectioning commands would help. The lack of concluding section at the end is also pretty unconventional.

Recommendation

Reject

  • validity: ok
  • significance: low
  • originality: low
  • clarity: poor
  • formatting: below threshold
  • grammar: acceptable

Author:  Abhijit Gadde  on 2025-12-03  [id 6101]

(in reply to Report 1 on 2025-11-27)

The main result of our paper is construction of an infinite family of pure state entanglement monotones that are easily computable. To the best of our knowledge, such monotones had not been constructed before. 
We could have considered a particularly simple family of monotones and proved its concavity in the partial trace; however, the aim of the paper is also to understand structural aspects of concavity thereby building a foundation for construction of new monotones. We did this by formulating concavity of the monotone as a condition of edge-convexity on the graph and proving a result such as the box-product of edge-convex graphs is edge-convex. Because of this broader goal, the paper is somewhat mathematical. We welcome referee's remarks regarding improving the readability. 
However, we do not believe that "the results do not make a contribution to scholarship significant enough to reach the level of publication in prominent journals like SciPost". We believe that our work is original and makes a solid contribution to the field.

---

## Editorial Decision

in_refereeing